# In-Context Transfer Learning: Demonstration Synthesis by Transferring Similar Tasks

## Abstract

In-context learning (ICL) is an effective approach to help large language models (LLMs) adapt to various tasks by providing demonstrations of the target task. Considering the high cost of labeling demonstrations, many methods propose synthesizing demonstrations from scratch using LLMs. However, the quality of the demonstrations synthesized from scratch is limited by the capabilities and knowledge of LLMs. To address this, inspired by transfer learning, we propose In-Context Transfer Learning (ICTL), which synthesizes target task demonstrations by transferring labeled demonstrations from similar source tasks. ICTL consists of two steps: source sampling and target transfer. First, we define an optimization objective, which minimizes transfer error to sample source demonstrations similar to the target task. Then, we employ LLMs to transfer the sampled source demonstrations to the target task, matching the definition and format of the target task. Experiments on Super-NI show that ICTL outperforms synthesis from scratch by 2.0% on average, demonstrating the effectiveness of our method.

## 1 Introduction

In-context learning (ICL) is an effective approach for large language models (LLMs) to adapt to various tasks based on the brilliant generalize ability of LLMs (Xun et al., 2017; Song et al., 2023b; Luo et al., 2024a). During the inference with ICL, input not only includes user questions but also several demonstrations to guide LLMs in generating answers correctly. Considering the high cost of demonstration labeling, many methods utilize LLMs to synthesize demonstrations from scratch without human involvement (Kim et al., 2022; Jin & Lu, 2024). For instance, Self-ICL (Chen et al., 2023b) employs LLMs to synthesize demonstration based on the task definition, while Su et al. (2024) improves the synthesis through iterations, where each iteration uses the previous results.

However, the synthesis using LLMs from scratch is constrained by the capabilities and knowledge of LLMs, limiting the quality of the synthesized demonstrations (Yu et al., 2023). For example, a model trained pre-2023 can not use knowledge after 2023, while a model not trained on coding tasks cannot understand code well (Rozière et al., 2024; Luo et al., 2024b). To solve this issue, thereby improving ICL performance while reducing human involvement, motivated by transfer learning (Pan & Yang, 2010; Iman et al., 2023), we ***propose to synthesize demonstrations for the target task by transferring the labeled demonstrations of similar tasks***. We use the idea of transfer learning since the previous works show that given similar source tasks, the performance of the target task can be enhanced according to the source task learning (Sun et al., 2020; Wang et al., 2024b). For example, as shown in Figure 1, the model can combine the *context* and the *answer* in the input of the sampled source demonstration, which is then used as the demonstration of the target task.

Based on the above discussion, we present **I**n-**C**ontext **T**ransfer **L**earning (ICTL), which obtains the demonstrations of the target task by transferring the demonstrations of the source tasks. ICTL consists of two steps: ***sample*** the demonstrations similar to the target task, and ***transfer*** the sampled demonstrations to the target task, as shown in Figure 1. First, we present an optimization objective to measure the transfer error, where we minimize the transfer error to sample the demonstrations highly similar to the target task. Then, we transfer the sampled demonstrations to the target task with LLMs, taking the sampled results and the target task definition as the input.

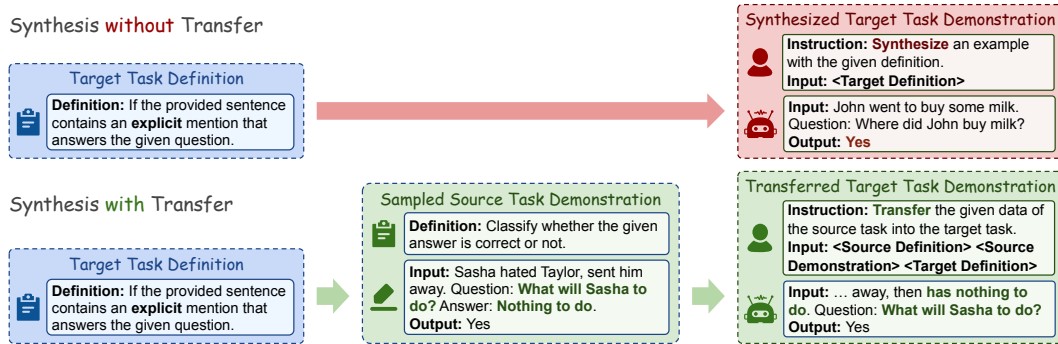

Figure 1: Comparison between previous demonstration synthesis methods (top) and our method (bottom). The blue part denotes the definition of the target task. The previous method synthesizes demonstration from scratch, while the model misinterprets the definition and generates a demonstration with the wrong answer, where the answer is not *explicit* mentioned by the sentence. In contrast, our method synthesizes demonstrations by transferring the sampled demonstrations, reducing the reliance on the capabilities of LLMs. The corresponding parts between the source and the target demonstrations of our method are marked in **bold**.

To validate ICTL, we conduct experiments on Super-NaturalInstructions (Super-NI) (Wang et al., 2022), which can fully evaluate the multi-task capability of models with more than $1,600$ different tasks. Compared to the demonstration synthesis by LLMs from scratch, our method achieves an average $2.0\%$ performance improvement, demonstrating its effectiveness. Further analysis shows that our method can effectively sample demonstrations that are highly similar to the target task from source tasks, showing the effectiveness of our optimization objective.

Our contributions are as follows:

- We argue that answering from scratch is constrained by the capabilities and knowledge of LLMs and thus propose synthesizing demonstrations by transferring labeled demonstrations of similar tasks;
- We introduce an optimization objective to guide the source sampling, ensuring the similarity between the sampled results and the target task;
- Experiments on Super-NI show that, compared with the synthesis from scratch, ICTL delivers a $2.0\%$ performance improvement on Super-NI, proving the effectiveness of ICTL.

## 2 RELATED WORKS

### 2.1 DEMONSTRATION SYNTHESIS

Demonstrations are of great importance in ICL, which can effectively help LLMs adapt various target tasks (Dong et al., 2024). Considering the high cost of human labeling, many methods present to synthesize demonstrations using LLMs from scratch, lowering the human involvement (Kim et al., 2022; Chang & Fosler-Lussier, 2023; Jin & Lu, 2024). Some methods focus on ensuring the correctness of the synthesized demonstrations, meeting the task definitions by filtering out low-quality synthesized results (Chen et al., 2023b; Su et al., 2024; Yang et al., 2024). Another type of method aims to increase the diversity of the synthesized demonstrations, creating ones dissimilar to synthesized results (Zhang et al., 2023; Shum et al., 2023; Wang et al., 2024a).

However, the demonstrations synthesized by the current methods are constrained by the knowledge and capabilities of LLMs themselves, limiting their performance on the tasks unseen in their pre-training (Yu et al., 2023). Although human-labeled demonstrations for new task scenarios can help LLMs generalize to these new tasks, labeling demonstrations for any new task or domain is costly (Wang et al., 2013). To address these issues, we present ICTL, which synthesizes demonstrations for new target scenarios by transferring labeled source demonstrations similar to the target task, addressing the limitation of the knowledge and capabilities of LLMs.

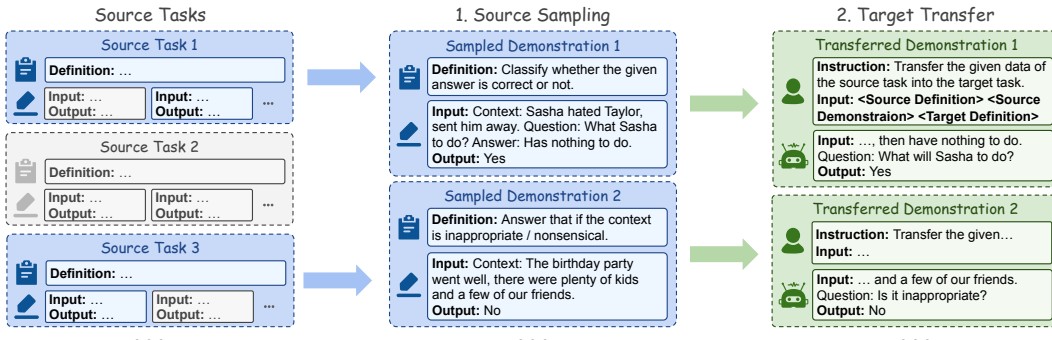

Figure 2: The illustration of ICTL, taking the target task definition "*If the provided sentence contains an explicit mention that answers the given question*" as an example. ICTL consists of two steps: *(i)* Source Sampling: sample demonstrations that are similar to the target task from the source tasks; *(ii)* Target Transfer: transfer the sampled demonstrations to the target task. The blue part indicates the task definitions and demonstrations similar to the target task, and the gray part indicates that it is dissimilar. The green part denotes the transferred demonstrations.

## 2.2 DEEP TRANSFER LEARNING

Transfer learning is a widely researched direction aimed at helping models acquire the ability to solve target tasks based on their existing capabilities from the source tasks (Pan & Yang, 2010; Zhuang et al., 2020). With the impressive performance demonstrated by deep learning methods, deep transfer learning has become an important approach within the field of transfer learning (Iman et al., 2023). Some methods focus on transferring and freezing model parameters to retain and learn features of different tasks (Scialom et al., 2022; Song et al., 2023a; Wang et al., 2023; Rostami et al., 2023; Du et al., 2024). Other transfer learning methods enhance the performance from the data perspective, studying how to adjust the training sequence of tasks, mix source task data with target task data, or modify the source task format to improve transfer learning performance (Xu et al., 2023; Wang et al., 2024b; Madine, 2024).

However, current transfer learning methods rely on the labeled data of the target task and the model training, leading to the high cost of the adaption considering the high cost of labeling and LLM training. Therefore, in this paper, we present to employ transfer learning to enhance ICL by synthesizing demonstrations using the labeled source demonstrations, lowering the human involvement and training cost, meanwhile helping LLMs adapt to various target tasks.

## 3 METHODOLOGY

In this section, we present ICTL, which synthesizes the demonstrations of the target task by transferring the labeled source demonstrations. The illustration of ICTL is shown in Figure 2, which consists of two steps: source sampling (§3.1) and target transfer (§3.2). Following the previous methods (Wang et al., 2024a; Yang et al., 2024), we synthesize demonstrations for each target task offline, where we do not synthesize for each target question since we want to ensure high efficiency of the inference. The prompts we used can be seen in Appendix B. The computational efficiency analysis of ICTL is shown in Appendix E.

## 3.1 SOURCE SAMPLING

The source sampling step is designed to sample demonstrations that are highly similar to the target task from the labeled source demonstrations. In this paper, we define the similarity as: If we want to sample $N$ source demonstrations, the $N$ source task demonstrations can minimize the target task error after transferring. We first present an optimization objective to guide the source demonstration sampling by minimizing the transfer error. Then, we discuss how to sample the source demonstrations similar to the target task using our objective specifically.

### 3.1.1 OPTIMIZATION OBJECTIVE FOR SOURCE SAMPLE

Supposing $S$ and $T$ represent the source and target tasks, respectively. $\epsilon(h)$ denotes the task error of the hypothesis $h$, $\hat{\mu}$ represents the empirical distribution for each task, $W$ is the Wasserstein distance (Rabin et al., 2012) measuring the divergence between two distributions, $N$ denotes the sample scale for each task, and $\varphi$ is a negligible function. The previous work (Redko et al., 2017) proves that the error of the transfer learning satisfies:

$$\epsilon_T(h) \leq \epsilon_S(h) + W(\hat{\mu}_S, \hat{\mu}_T) + \varphi(N_S, N_T) \tag{1}$$

Further details of Equation 1 are discussed in Appendix A. From Equation 1, we can see that the upper bound of the error for the target task is mainly determined by the error of the source task and the divergence between the source and target tasks. It is hard to reduce the source task error since the source demonstrations can not be modified. So we aim to minimize the target error by minimizing the divergence between the source and target tasks $W(\hat{\mu}_S, \hat{\mu}_T)$.

However, directly minimizing the upper bound results in $\hat{\mu}_T = \hat{\mu}_S$, which makes the transferred demonstrations irrelevant to the target task. Therefore, giving $x$ as the representation vector of the task definition, we ask $\hat{\mu}_T$ to satisfy that:

$$\hat{\mu}_T = \arg\min_{\hat{\mu}} W(\hat{\mu}, \hat{\mu}_S) + W(\hat{\mu}, x_T) \tag{2}$$

In Equation 2, the first term minimizes the divergence between the target and source demonstrations, and the second term ensures that the target demonstrations are consistent with the target task definition. When calculating the Wasserstein distance, if an input is a point (vector), we regard it as a distribution with a variance of $0$. We discuss the effectiveness of Equation 2 with experiments in Appendix F.2.

Given a series of source tasks $\{S_i\}$, suppose $N$ is the sampling scale of demonstrations from multiple source tasks $\{\hat{\mu}_{S_i}\}$, $N_{S_i}$ is the sampled number of $S_i$ and $\hat{\mu}$ is the empirical distribution of all possible sampled source demonstrations. Based on Equation 1 and Equation 2, we can derive the optimization objective to sample the source demonstrations:

$$\hat{\mu}_S = \arg\min_{\hat{\mu}} \sum_{S_i} \frac{N_{S_i}}{N} (6W(\hat{\mu}_{S_i}, x_T) + W(x_{S_i}, x_T)) \tag{3}$$

The proof of Equation 3 is provided in Appendix A. It can be observed that the first term in the summation ensures that the sampled source task demonstrations are similar to the target task definition, and the second term ensures that the source task definitions are similar to the target task definition. Using Equation 3, we can sample source demonstrations highly similar to the target task, thereby lowering the transfer error, and ensuring the quality of the transferred demonstrations.

### 3.1.2 SAMPLING WITH EQUATION 3

Based on the above discussion, we then discuss how to sample source demonstrations specifically. First, we embed the definitions and demonstrations of all source tasks, as well as the definition of the target task, into vectors using an embedding model. Following previous work (Wang et al., 2024b), we then filter the source tasks to select those most similar to the target task, reducing the overhead of subsequent calculations while ensuring performance. The filtering is done by ranking the Wasserstein distance between the embedding vectors of the source and target task definitions. From the filtered source tasks, we sample a fixed number of demonstrations using Equation 3. We employ a randomized algorithm for the sampling, with details provided in the Appendix C.

### 3.2 TARGET TRANSFER

The target transfer step focuses on transferring the sampled demonstrations to the target task while ensuring that the transferred demonstrations are consistent with both the target task and the sampled demonstrations, transcending the limitations of the inherent capabilities and knowledge of LLMs. The target transfer step consists of: *Transfer*, *Verify*, and *Sample*.

**Transfer** is to transfer the sampled demonstrations to match the target task definition and format. We employ LLMs for the transfer, where the input includes the definitions of both the source and target tasks, the source demonstration to be transferred, and a human-labeled example of the target task to specify the input and output formats.

**Verify** is designed to check whether the transferred demonstration is consistent with the definition of the target task, improving the quality of the transferred demonstrations. We employ LLMs to verify the transferred results. The target task definition, one example, and the transferred demonstration are provided as input to check whether the transferred demonstration consistent with the task definition, with the correct input and output formats. Any demonstration verified by the LLM as inconsistent is discarded to ensure the quality of the transferred results.

**Sample** is to sample the verified target demonstrations with Equation 2, ensuring that the sampled demonstration is consistent with the target task while staying similar to the sampled source demonstrations, thereby transcending the limitations of the capabilities and knowledge of LLMs. The sampling algorithm used for the transferred demonstration sampling is the same as the source sampling, with the optimization objective defined by Equation 2. The sampled demonstrations are considered as the final output of our transfer method.

## 4 EXPERIMENTS

### 4.1 EXPERIMENT SETUP

#### 4.1.1 DATASET

We use the Super-NaturalInstructions dataset (Super-NI) (Wang et al., 2022) to validate our method, which contains over $1,600$ tasks, allowing for a comprehensive evaluation of the model cross-task generalization ability. Following previous work (Wang et al., 2024b), we conduct experiments on all English tasks in Super-NI, including $756$ tasks in the training set and $116$ tasks in the test set. Based on prior research (Wang et al., 2024b), we categorize all tasks in the test set into six categories to better analyze the performance of our method across different tasks, as shown in Appendix D.

#### 4.1.2 METRIC

Following the Super-NI setup, we use Rouge-L (RougeL) and Exact Match (EM) as the evaluation metrics. RougeL measures the overlap between the predicted output and the reference answer, while EM assesses whether the predicted output exactly matches the reference. Following Wang et al. (2022), we mainly use RougeL as the evaluation metric, since EM is not suitable for tasks that can be answered in multiple ways (e.g., summarization, title generation).

#### 4.1.3 MODEL

We use BGE-EN-ICL (Chen et al., 2023a) to embed task definition and demonstrations for the sampling, which is the state-of-the-art (SOTA) embedding model during our experiments. For the transfer and inference, we use Llama3.1-8b-Instruct (Llama3.1-8b) (Dubey et al., 2024) and `GPT-4o` (OpenAI et al., 2024) as the experimental models. Llama3.1-8b is one of the current best-performing open-source LLMs. `GPT-4o` is one of the most powerful LLMs at present, which achieves SOTA performance on multiple mainstream benchmarks. We mainly use Llama3.1-8b as the model of our analysis experiments due to the high cost of `GPT-4o`.

#### 4.1.4 BASELINE

To thoroughly evaluate the effectiveness, we compare ICTL with the following baselines:

- **Zero**: No demonstrations are provided during inference, using a zero-shot setting;
- **Direct**: Directly use the sampled source demonstrations without transferring;
- **Single**: Only use the single human-labeled example as the demonstration;
- **Synthesis**: Synthesize demonstrations from scratch based on the one example provided.

Table 1: The main experiment results on Super-NI. For each category, we use RougeL for evaluation. The best result for each category is highlighted in **bold**. Considering the high cost of GPT-4o, we only adapt experiments on 12 tasks of the Super-NI test set for GPT-4o, where we randomly select 2 tasks for each category, as shown in Appendix D.

| Model | Category | Zero | Direct | Single | Synthesis | ICTL |
|---|---|---|---|---|---|---|
| **Llama3.1-8b** | Classification | 62.5 | 60.3 | 61.9 | 65.4 | **68.0** |
| | Comprehension | 56.1 | 55.3 | 60.0 | 62.8 | **67.8** |
| | Dialogue | 57.2 | 62.7 | 65.2 | **73.1** | 72.3 |
| | Extraction | 43.4 | 38.7 | 48.3 | **53.2** | 51.2 |
| | Generation | 38.4 | 34.6 | 41.1 | 42.3 | **45.8** |
| | Rewriting | 46.6 | 32.6 | 58.1 | 60.5 | **61.0** |
| | Overall (EM) | 36.9 | 35.6 | 39.7 | 41.9 | **44.0** |
| | Overall (RougeL) | 52.0 | 48.8 | 54.7 | 57.8 | **60.3** |
| **GPT-4o** | Classification | 76.0 | 72.2 | 78.0 | 79.0 | **81.0** |
| | Comprehension | 78.4 | 76.4 | 74.9 | 72.2 | **78.4** |
| | Dialogue | 80.5 | 78.5 | 80.5 | **83.5** | 82.0 |
| | Extraction | 72.7 | 65.2 | **73.0** | 71.0 | 70.9 |
| | Generation | 39.1 | 38.4 | 42.6 | 44.5 | **45.4** |
| | Rewriting | 65.3 | 59.3 | 79.6 | 80.2 | **80.7** |
| | Overall (EM) | 49.2 | 44.6 | 49.4 | 49.7 | **51.8** |
| | Overall (RougeL) | 68.7 | 65.0 | 71.4 | 71.8 | **73.1** |

### 4.1.5 Implementation Detail

During sampling, we first select 16 source tasks that are most similar to each target task. For each target task, we sample 128 demonstrations from the source tasks to be transferred. Since Super-NI labels more than one answer for some questions, we transfer each answer with the question separately. For the transferred results, we sample 512 demonstrations for the inference. We employ the 3-shot inference, selecting demonstrations for each test question based on the BM-25 similarity. The reason for the parameter selection in this part is discussed in §4.4.

### 4.2 Main Experiment

As shown in Table 1, ICTL outperforms all baselines without transfer across different metrics and models on most categories, showing the effectiveness of our method. Additionally, the results in the table also reveal the following insights:

**Baseline** Compared to all baselines without transfer, our method achieves better performance, demonstrating the effectiveness of transferring. Notably, ICTL brings 2.0% improvement on average compared to the *Synthesis* setting. This shows that the demonstrations synthesized by LLMs from scratch are constrained by the capabilities and knowledge of LLMs themselves. In contrast, ICTL overcomes this constraint by providing the labeled demonstrations of other similar tasks, lowering the capability and knowledge requirement. Additionally, the *Direct* setting directly using the sampled results as demonstrations leads to worse performance compared to the *Zero* setting. This indicates that transfer is necessary when using demonstrations from other tasks to enhance performance, even if the sampled source demonstrations are highly similar to the target task.

**Task** ICTL improves performance across most task categories, proving its effectiveness. Specifically, the performance improvement is more significant for tasks with a higher rate in all test data, as there are sufficient similar source demonstrations for transfer, where the rates of different tasks are shown in Appendix D. However, our method slightly underperforms compared to other settings in the *Dialogue* and *Extraction* tasks. This is because these two tasks comprise only about 5% of the total data, leading to lower-quality transfer results due to a lack of similar source demonstrations. These findings suggest that it is important to use source demonstrations that are highly similar to the target task, as discussed in detail in §4.4.3. To better observe the relationship between the source and target tasks of various categories, we static the transfer status of ICTL in Appendix F.3.

**Metric** On both the EM and RougeL metrics, ICTL results in performance improvements, demonstrating its effectiveness. Compared to EM, the performance improvement on RougeL is more significant. That is because EM is harder to improve since it requires the generated answer to be completely identical to the reference answer, while RougeL allows for partial matches and flexibility in answer formats, providing credit for partially correct outputs, making it relatively easier to improve.

**Model** With both Llama3.1-8b and `GPT-4o`, ICTL demonstrates performance improvements, confirming its effectiveness on LLMs with different levels. Besides, compared to Llama3.1-8b, the performance enhancement of `GPT-4o` is somewhat weaker. That is because, it can be observed that even under the *Zero* setting without demonstrations, `GPT-4o` is already capable of effectively addressing the tasks within Super-NI. Therefore, when the model struggles to adequately tackle the target task on itself, ICTL can yield more significant performance gains.

### 4.3 ABLATION STUDY

To verify the effectiveness of each component in ICTL, we conduct ablation studies, where the experimental results are shown in Table 2. Based on the table, we analyze each ablation study in order of its impact on performance, from most to least significant.

Table 2: The ablation experiment results using Llama3.1-8b for the following components: *(i)* Transfer Verify: remove target verification; *(ii)* Source Sample: sample source demonstrations randomly; *(iii)* Target Sample: directly use the verified target demonstrations without sampling.

| Method | EM | RougeL |
|---|---|---|
| ICTL | 44.0 | 60.3 |
| - Target Verify | 41.4(−2.6) | 56.3(−4.0) |
| - Source Sample | 41.7(−2.3) | 56.8(−3.5) |
| - Target Sample | 43.7(−0.3) | 60.0(−0.3) |

**Target Verify** Removing transfer verification results in the most significant performance drop of 3.3% on average across two metrics. This indicates that the quality of demonstrations transferred directly is relatively low, showing the necessity of the verification. There are two main reasons for the low quality of demonstrations transferred directly: *(i)* For many test tasks, especially those that can be answered in multiple ways, it is difficult for LLMs to determine the format of the task, resulting in poor transfer results; *(ii)* Previous research (Min et al., 2022) shows that LLMs could generate responses according to their prior experience during the pre-training while ignoring instructions, resulting in some generated results not meeting the definition and format of the target task.

**Source Sample** Removing source sampling also causes a sharp performance drop of 2.9% on average. This is because, without source sampling, our method uses random sampling of source demonstrations, which leads to many dissimilar source demonstrations being sampled, decreasing the performance. This result proves the necessity of sampling the source demonstrations according to the similarity to the target task before the transfer. Besides, after removing source sampling, the performance of ICTL is near the *Synthesis* setting. This shows that when the source demonstrations provided are significantly different from the target task, LLMs are more inclined to synthesize results by themselves without referring to the demonstrations provided.

**Target Sample** Removing target sampling has the least impact on performance, causing only a 0.3% decrease. This is because, considering that ensuring the similarity between the demonstration and the question can effectively ensure the performance of ICL (Shum et al., 2023; Yang et al., 2024), during the evaluation, we also select the demonstration corresponding to each question based on BM-25, which overlaps with transfer sampling to a certain extent.

### 4.4 ANALYSIS

In this part, we analyze how different parameters affect the performance of ICTL to guide the selection of parameters in practical applications, as shown in Figure 3. To better observe the performance changes brought about by ICTL with the change of different parameters, we use the *Single* setting as our baseline. We also present the case study in Appendix G to present how ICTL transfer demon-

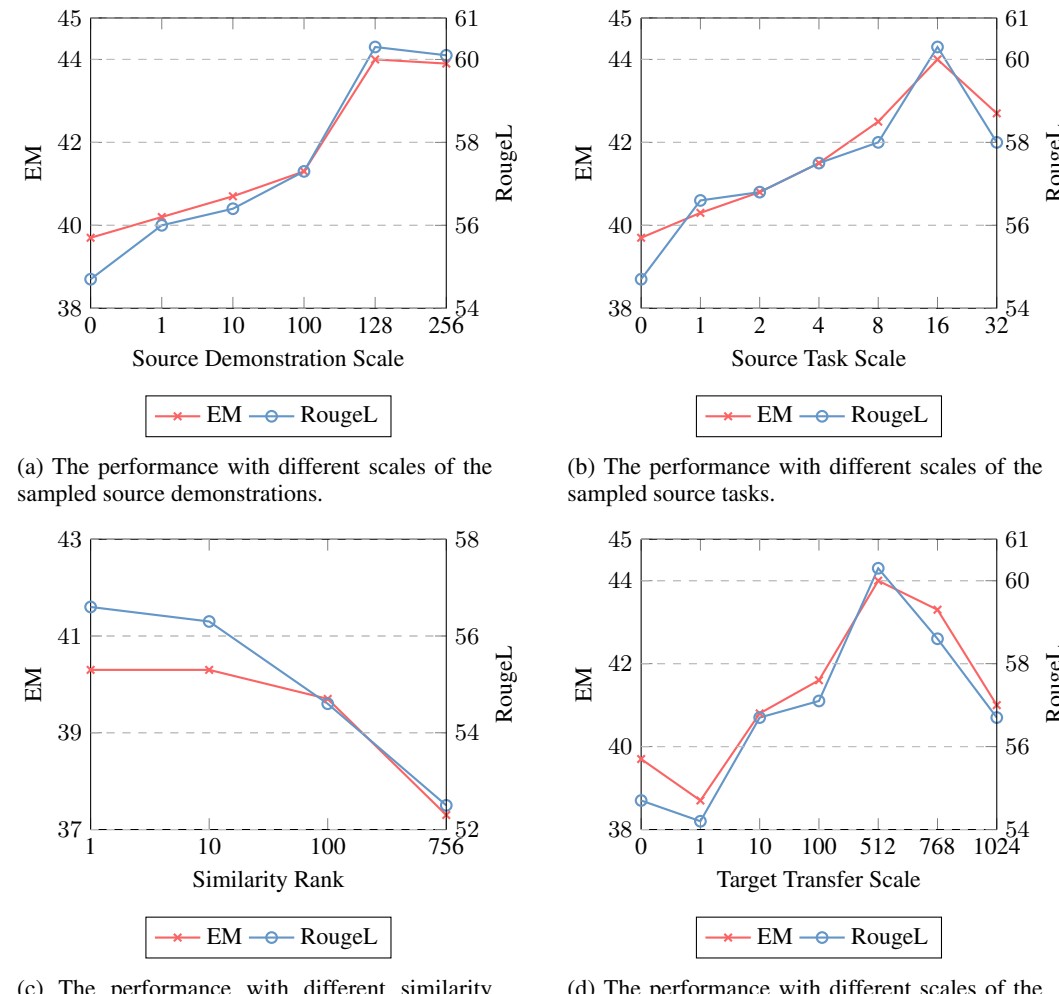

(a) The performance with different scales of the sampled source demonstrations.

(b) The performance with different scales of the sampled source tasks.

(c) The performance with different similarity ranks to the target task.

(d) The performance with different scales of the transferred target demonstrations.

Figure 3: The impact of different parameters on the performance of the Super-NI test set with ICTL using Llama3.1-8b. 0 of the X-axis indicates the performance under the *Single* setting.

strations, and evaluate the performance of ICTL under human-labeled target task demonstrations and cross-domain settings in Appendix F.5 and Appendix F.6.

### 4.4.1 SOURCE DEMONSTRATION SCALE

The scale of source demonstrations available for different practical applications varies, so we analyze the impact of different scales of source demonstrations on the performance of our method, as shown in Figure 3a. From the figure, we can see that: *(i)* When the scale of the source demonstration sampling is smaller than 128, the overall experimental results exhibit an upward trend, demonstrating that increasing the amount of source demonstrations can effectively enhance the performance of our method; *(ii)* When the sampling scale exceeds 128, there is a slight decrease in performance, indicating that further addition of new source demonstrations does not continue to improve performance, as the number of demonstrations similar to the target task is limited. Therefore, when obtaining demonstrations of source tasks, it is necessary to obtain as many demonstrations as possible to ensure that there are enough different abilities or knowledge for the target task.

Notably, compared to not using transfer learning, even transferring using one source demonstration can also effectively improve the performance the target task. This is because: *(i)* Even using one single source demonstration, we can also synthesize a large amount demonstrations of the target

task, resulting in a high-quality demonstration pool and thus better performance than without transfer learning; *(ii)* Previous research (Kim et al., 2022; Wang et al., 2024a) and the *Synthesis* setting of Table 1 show that even without source demonstrations, LLMs can still synthesize demonstrations based on the inherent knowledge of themselves, thereby enhancing inference performance.

### 4.4.2 SOURCE TASK SCALE

The scale of source tasks that can be obtained varies in practical applications, so we analyze the impact of different task scales on the performance of ICTL. The experimental results are shown in Figure 3b, from which we can see that: *(i)* When the scale of source tasks is less than 16, the overall performance exhibits an upward trend, while when the scale exceeds 16, the performance starts to decline sharply, showing that blindly increasing the scale of the source task cannot bring about continuous improvement and the importance of ensuring the similarity between the source and the target tasks; *(ii)* Compared to the source demonstration scale, the performance degradation is more pronounced with the increase in source task scale, since the scale of source tasks similar to the target task is limited, whereas simply increasing the scale of tasks, rather than the demonstrations, introduces more irrelevant information, leading to a more significant decrease in the quality of the transferred demonstrations and the inference performance.

### 4.4.3 TASK SIMILARITY RANK

Considering there could be many new tasks emerging in future research and applications, to explore the adaptability of ICTL to new tasks, we conduct experiments to examine the impact of the similarity between the source and target tasks on performance. We rank the Wasserstein distance of the embedding vectors of the source and target task definition in descending order, selecting the 1st, 10th, 100th, and last-ranked (756th in the Super-NI train set) source tasks to be transferred. The experimental results are shown in Figure 3c, from which we can observe the following: *(i)* When the similarity ranking of the source tasks is within the top 10, the performance of our method does not fluctuate significantly, since there exists multiple source tasks similar to those in the Super-NI test set, resulting in transferred demonstrations of comparable quality; *(ii)* After the similarity ranking exceeds 10, the performance of our method begins to decline sharply, indicating that demonstrations of tasks with large gaps can not help the target task, showing the importance of ensuring the similarity between the source tasks and target tasks.

### 4.4.4 TARGET TRANSFER SCALE

Due to the computational resource limitation in practical applications, the scale of the transferred demonstrations could be limited. Therefore, we evaluate the performance of ICTL under different scales of transferred demonstrations, as shown in Figure 3d. From the figure, we can observe the following: *(i)* In cases where only one single demonstration is transferred, the model performance decreases compared to without transfer, since the quality of the single transferred demonstration is lower than the provided example labeled by humans, leading to a performance decline; *(ii)* Even only obtains 10 demonstrations by transferring, our method achieves better performance than no transfer, whereas the scale of transferred demonstrations increases, the performance improves accordingly, demonstrating the necessity of sufficient transferring; *(iii)* However, after the transferred demonstrations reach a certain scale, the model performance plateaus, since the information contained in the sampled source demonstrations is fully represented with 512 transferred demonstrations, and further increasing the scale does not yield new high-quality demonstrations, while the performance is reduced since mixing more low-quality demonstrations.

## 5 CONCLUSION

In this paper, motivated by transfer learning, we propose ICTL, which synthesizes the demonstrations of the target task by transferring the similar labeled demonstrations, addressing the constraint that synthesizing from scratch with LLMs is limited by the capabilities and knowledge of LLMs. We first present an optimization objective for sampling source demonstrations, aiming to minimize transfer errors by ensuring that sampled demonstrations are highly similar to the target task. Subsequently, we transfer the sampled demonstrations to the target task using LLMs without human

involvement, taking the sampled results and the target task definition as the input. Experiments on Super-NI demonstrate that our method achieves an average improvement of 2.0% over demonstrations synthesized without transfer, validating its effectiveness. Additionally, analysis confirms that our method ensures a high similarity between sampled source demonstrations and the target task, proving the effectiveness of our proposed optimization objective.

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

## A    PROVE OF EQUATION 3

In this section, we present the proof of Equation 3. The proof includes three parts. First, we discuss how to measure the transfer error when transferring across multiple source tasks. Next, we address how to measure the discrepancy between the source tasks and the target task, denoted as $W(\hat{\mu}_S, \hat{\mu}_T)$. Finally, we combine the existing results to derive Equation 3.

$$\epsilon_T(\hat{h}_{\boldsymbol{\alpha}}) \leq \min_h \epsilon_T(h) + c_1 + 2\sum_{i=1}^N \alpha_i \left( W(\hat{\mu}_{S_i}, \hat{\mu}_T) + \lambda_i + c_2 \right) \tag{4}$$

Suppose $\boldsymbol{\alpha} = \{\alpha_i\}$ represents the proportion of each source task, $c_1, c_2$ are dependent on $n, N_{S_i}, N_T$, and $\lambda_i = \min_h(\epsilon_{S_i}(h) + \epsilon_T(h))$ denotes the joint error of each source task $S_i$. Based on Equation 1, the previous work (Redko et al., 2017) has proved that, for the transfer learning across multiple source tasks, the error satisfies Equation 4.

$$\hat{\mu}_S = \underset{\{\hat{\mu}_{S_i}\}_N}{\arg\min} \sum_{i=1}^N \alpha_i W(\hat{\mu}_{S_i}, \hat{\mu}_T) \tag{5}$$

To minimize the error, we aim to minimize the upper bound of the error. Since $\min_h \epsilon_T(h) \leq \sum_{i=1}^N \alpha_i \epsilon_T(h_{S_i})$, and $\sum_{i=1}^N \alpha_i \lambda_i \leq \sum_{i=1}^N \alpha_i \epsilon_T(h_{S_i}) + \alpha_i \epsilon_{S_i}(h)$, by replace $\epsilon_T(h_{S_i})$ with Equation 1, and ignoring the terms related to the error of source tasks and constants unrelated to $\mu$, we can obtain Equation 5. Equation 1 defines how to sample the target demonstrations given the source demonstrations. Then, we discuss the upper bound of the value of Equation 1, where we can adjust the source demonstrations to minimize the upper bound, thereby lowering the transfer error.

**Theorem 1** *Let $x_S, x_T$ represent the representation vectors of the task definition of $S$ and $T$. If*

$$\hat{\mu}_T = \underset{\hat{\mu}}{\arg\min}\, W(\hat{\mu}, \hat{\mu}_S) + W(\hat{\mu}, x_T),$$

*then*

$$W(\hat{\mu}_S, \hat{\mu}_T) \leq 6W(\hat{\mu}_S, x_T) + W(x_S, x_T).$$

**Proof 1** *Let $\hat{\mu}_{S,T}$ represent the empirical distribution of the subset sampled from $X_S$, which has the data most close to $x_T$. It is obvious that $W(\hat{\mu}_{S,T}, x_T) \leq W(\hat{\mu}_S, x_T)$.*

*Because $\hat{\mu}_T = \arg\min_{\hat{\mu}} W(\hat{\mu}, \hat{\mu}_S) + W(\hat{\mu}, x_T)$, we can get:*

$$\begin{aligned}
W(\hat{\mu}_T, \hat{\mu}_S) + W(\hat{\mu}_T, x_T) &\leq W(\hat{\mu}_{S,T}, \hat{\mu}_S) + W(\hat{\mu}_{S,T}, x_T) \\
&\leq W(\hat{\mu}_{S,T}, \hat{\mu}_S) + W(\hat{\mu}_S, x_T) \\
&\leq W(\hat{\mu}_{S,T}, x_T) + 2W(\hat{\mu}_S, x_T) \\
&\leq W(\hat{\mu}_{S,T}, \hat{\mu}_T) + W(\hat{\mu}_T, x_T) + 2W(\hat{\mu}_S, x_T)
\end{aligned}$$

*Erase $W(\hat{\mu}_T, x_T)$ on both sides of the unequal sign, we can get:*

$$\begin{aligned}
W(\hat{\mu}_T, \hat{\mu}_S) &\leq W(\hat{\mu}_{S,T}, \hat{\mu}_T) + 2W(\hat{\mu}_S, x_T) \\
&\leq W(\hat{\mu}_{S,T}, x_T) + W(\hat{\mu}_T, x_T) + 2W(\hat{\mu}_S, x_T) \\
&\leq 3W(\hat{\mu}_S, x_T) + W(\hat{\mu}_T, x_T) + W(\hat{\mu}_T, \hat{\mu}_S) \\
&\leq 3W(\hat{\mu}_S, x_T) + W(\hat{\mu}_{S,T}, \hat{\mu}_S) + W(\hat{\mu}_{S,T}, x_T) \\
&\leq 5W(\hat{\mu}_S, x_T) + W(\hat{\mu}_{S,T}, x_T) + W(x_T, x_S) \\
&\leq 6W(\hat{\mu}_S, x_T) + W(x_T, x_S)
\end{aligned}$$

*Thus, we conclude:*

$$W(\hat{\mu}_T, \hat{\mu}_S) \leq 6W(\hat{\mu}_S, x_T) + W(x_T, x_S).$$

Theorem 1 provides an upper bound for measuring the difference between the demonstrations of the target task and the source task in task transfer, based on the discrepancy between the task definitions of the source and target tasks. The reason this measurement holds is that the demonstrations for the target task are entirely transferred from the source demonstrations and the target task definition, meaning they can describe its characteristics. By substituting Theorem 1 into Equation 5, we can derive Equation 3.

## B  PROMPTS OF ICTL

Table 3: The prompt of transfer.

---

**The Prompt of Transfer of ICTL**

---

Convert an example from Task A into an example for Task B, ensuring that both examples are consistent in terms of domain and knowledge. A sample for Task A is provided below. Please create a corresponding example for Task B, while maintaining the same domain and knowledge context.
The definition of Task A: {task_a_definition}
The definition of Task B: {task_b_definition}

—

For example, given the following example for Task A:
Input:
{task_A_question_demo}
Reason:
{task_A_rationale_demo}
Answer:
{task_A_answer_demo}

The corresponding example for Task B could be:
Input:
{task_B_question_demo}
Reason:
{task_B_rationale_demo}
Answer:
{task_B_answer_demo}

—

Based on the above example, please transfer the following example from Task A to Task B:
Input:
{task_A_question}
Answer:
{task_A_answer}

Your output format should be as follows:
Input:
<Converted input of Task B >
Reason:
<Explanation of the converted >
Answer:
<Converted answer of Task B >

---

The prompts we used in ICTL are shown in Table 3, Table 4 and Table 5.

## C  ALGORITHM FOR DATASET SAMPLING

In this section, we introduce the specific design of the randomized algorithm for sampling. The algorithm utilizes simulated annealing (Bertsimas & Tsitsiklis, 1993) to optimize the sampling of demonstrations most similar to the target task with low computational costs.

Table 4: The prompt of verification.

| The Prompt of Verification of ICTL |
| --- |
| Given a task description, several examples, and a pre-synthesized example, evaluate whether the pre-synthesized example matches the format and functionality of the provided examples and aligns with the task description. Based on the evaluation, determine whether the pre-synthesized example is "Qualified" |
| You should check the pre-synthesized example based on the following criteria: |
| 1. Format Consistency: Does the pre-synthesized example follow the format of the provided examples? |
| 2. Task Fulfillment: Does the pre-synthesized example fulfill the requirements of the task description? |
| 3. Functional Accuracy: Are the input and output in the pre-synthesized example consistent with those in the provided examples? |
| If the pre-synthesized example meets all the criteria above, return: "Qualified." |
| If the pre-synthesized example fails to meet any of the criteria, return: "Unqualified." |
| Think it step by step. |
| |
| Task Description: |
| {definition} |
| |
| Examples: |
| |
| Input: |
| {input_demo} |
| Reason: |
| {reason_demo} |
| Answer: |
| {answer_demo} |
| |
| — |
| |
| ... |
| |
| — |
| |
| Pre-synthesized Example: |
| Input: |
| {input_transferred} |
| Reason: |
| {reason_transferred} |
| Answer: |
| {answer_transferred} |

Simulated annealing is a probabilistic global optimization algorithm that initially accepts suboptimal solutions at high temperatures to avoid local optima. As the temperature gradually decreases, the algorithm converges. The initial solution is generated through random sampling, where samples from the given demonstrations are randomly selected as the starting candidate solution. We use Equation 3 and Equation 2 as the score function to evaluate the quality of random sampling from the given demonstrations, where we calculate the Wasserstein distance following Rostami et al. (2023).

During each iteration, the algorithm perturbs the current candidate solution to generate a new one. If the algorithm fails to find a better solution after several attempts, the perturbations are triggered to escape local optima. Whether the perturbed candidate is accepted depends on the difference in scores between the new and current solutions. Even if the new candidate is worse, there is a certain probability it is accepted. This probability decreases as the temperature drops, promoting sufficient search space exploration.

The annealing process starts with an initial temperature of $1.0$, with a cooling rate of $0.99$. The temperature decays after each iteration until it reaches the minimum value of $10^{-4}$, at which point the algorithm stops. Additionally, we set a threshold: if no better solution is found after $100$ iterations, large-step perturbations are applied. Although our method demands the additional cost for comput-

Table 5: The prompt of inference.

| The Prompt of Inference of ICTL |
|---|
| `{task_definition}` 
 Here are some demonstrations of the task: 

 — 

 Input: 
 `{input_demo}` 
 Reason: 
 `{reason_demo}` 
 Answer: 
 `{answer_demo}` 

 — 

 ... 

 — 

 Based on the above demonstrations, please generate a response to the following question. 
 Your output format should be as follows: 
 Reason: 
 <Explanation of the answer > 
 Answer: 
 <Your answer > 
 Think it step by step. 

 Input: 
 `{input_user}` |

ing simulated annealing compared with the general ICL methods, these costs are offline, where our method has the same inference cost as other general ICL methods.

## D   CATEGORY OF SUPER-NI TEST TASKS

Table 6: Category of the Super-NI test set. The tasks used for `GPT-4o` experiments are marked in **bold**.

| Category | Task ID |
|---|---|
| Classification | 20, 50, 190, 199, 200, 201, 202, 226, 232, 233, 242, 290, 349, 391, 392, 393, 520, 614, 623, 640, **641**, 642, 738, 827, 828, 890, 935, 936, 937, 970, 1344, 1385, 1386, 1387, 1388, 1393, 1439, 1442, 1516, **1529**, 1554, 1612, 1615, 1624, 1640 |
| Comprehension | **33**, 133, 249, 304, 329, 330, 401, **648**, 891, 892, 893, 1390, 1391, 1664 |
| Dialogue | 362, 879, 880, 1394, 1531, **1533**, **1534** |
| Extraction | 36, **39**, **281**, 613, 620, 645 |
| Generation | 102, 219, 220, 288, 418, 500, 510, 569, 602, 619, 677, 743, 760, 769, 957, 1152, **1153**, 1154, 1155, 1156, 1157, 1158, 1159, 1161, 1342, 1356, **1358**, 1407, 1409, 1540, 1586, 1598, 1631, 1659, 1728 |
| Rewriting | **34**, 35, 121, 402, 442, **670**, 671, 1195, 1345, 1557, 1562, 1622 |

The category of the Super-NI test set is shown in Table 6, where we follow the category of Wang et al. (2024b). To better observe the impact of demonstration volume on transfer performance, we also count the distribution of demonstrations corresponding to different categories of tasks in the Super-NI test set, as shown in Figure 4.

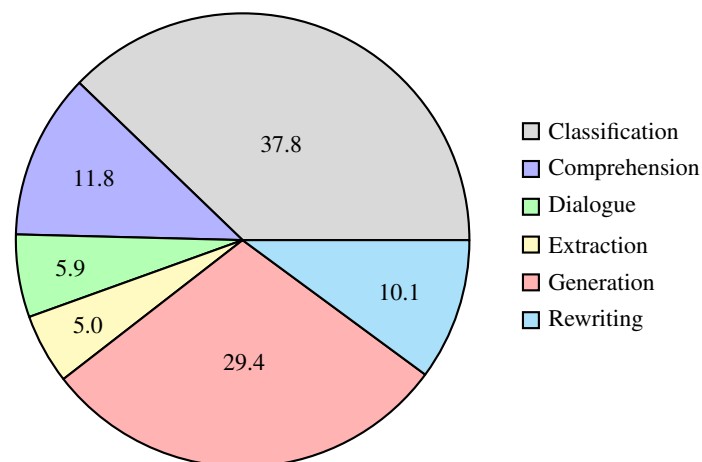

Figure 4: Category distribution of the Super-NI test set.

# E EFFICIENCY ANALYSIS OF ICTL

## E.1 EFFICIENCY OF DEMONSTRATION SYNTHESIS

In this section, we provide a detailed analysis of the computational efficiency of ICTL. Our goal is to analyze how the efficiency of source sampling and target transfer impacts the overall runtime and resource utilization, particularly in terms of the source demonstration scale and model inference time.

Let $N_s$ represent the total scale of the source demonstrations, $N_s^S$ the scale of the sampled source demonstrations, and $N_t^S$ the scale of the sampled target demonstrations. The symbol $c_\theta$ denotes the time taken by the sampling algorithm to process one single data with parameter $\theta$. Similarly, $c_\mathcal{M}$ represents the time for the model $\mathcal{M}$ to process a single data.

$$c_\theta N_s N_s^S + c_\mathcal{M} N_s^S + c_\mathcal{M} N_s^S + c_\theta N_s^S N_t^S \tag{6}$$

Then, we can represent the total computational cost with Equation 6. In Equation 6, the first term represents the efficiency of source sampling, the second term corresponds to the target transfer, the third term describes the transfer verification, and the fourth term reflects the efficiency of the sampling of the synthesized demonstrations.

$$(c_\theta N_s + 2c_\mathcal{M})N_s^S + c_\theta N_s^S N_t^S \tag{7}$$

Based on Equation 6, we can derive Equation 7. From the equation, it can be observed that the total runtime is primarily dependent on $N_s^S$, which is the scale of the sampled demonstrations. Therefore, when computational resources are limited and the overall scale of the source demonstrations $N_s$ is large or the model inference time $c_\mathcal{M}$ is high, we can reduce $N_s^S$ to improve efficiency.

## E.2 EFFICIENCY OF INFERENCE

| Setting | Zero | Direct | Single | Synthesis | ICTL |
|---|---|---|---|---|---|
| **Average Tokens** | 95.7 | 257.3 | 156.7 | 278.7 | 262.3 |

Table 7: The average input token number during inference under different settings on Super-NI.

To evaluate the efficiency of ICTL during inference, we calculate the average input token numbers under different settings, as shown in Table 7. From the table, we can see that, during inference,

the average token number of our method is similar to Direct and Synthesis. This is because, the demonstration generation is offline, where during the inference, we only need to sample question-related demonstrations from the generation results, having a similar efficiency to the general ICL methods.

## F  FURTHER ANALYSIS EXPERIMENT

### F.1  PERFORMANCE OF DIFFERENT SOURCE SAMPLING METHODS

| Retriever | Direct | ICTL |
|---|---|---|
| BM25 Robertson & Zaragoza (2009) | 46.2 | 55.8 |
| Contriever Lei et al. (2023) | 46.5 | 56.3 |
| Dr.ICL Luo et al. (2023) | 48.4 | 58.7 |
| ICTL | **48.8** | **60.3** |

Table 8: The RougeL of ICTL filtering source task data with different retrieval methods under two settings (Direct, ICTL) on Super-NI using Llama3.1-8b. The best performance is marked in **bold**.

To further prove the effectiveness of ICTL, we compared the demonstration transfer performance using different source task sampling methods. The experimental results are shown in Table 8, where we can see that the sampling method of ICTL is better than other sampling methods, proving the effectiveness of ICTL.

### F.2  TARGET SAMPLING DIVERGENCE

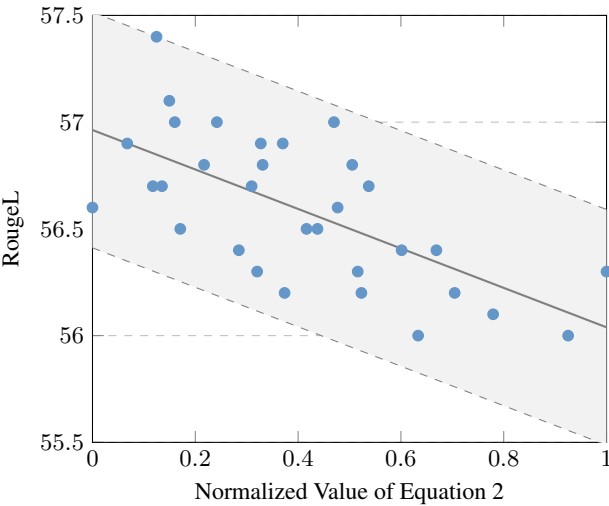

Figure 5: RougeL on the Super-NI test set using the 32 different sets of randomly sampled transferred demonstrations with different values of Equation 2 using Llama3.1-8b. To better observe the changes, we normalize the values of the X-axis.

To validate the effectiveness of Equation 2 as a sampling metric, we randomly sample 32 different sets of synthesized demonstrations. For each set, 128 demonstrations are randomly selected for each task, where the corresponding Equation 2 values and performance are shown in Figure 5. From the figure, we can observe the following: *(i)* As the Equation 2 value increases, the model performance shows a declining trend, indicating that the equation we proposed can effectively evaluate the divergence between the source demonstrations, the target task definition, and the synthesized demonstrations, which in turn helps assess model performance; *(ii)* The variation in all experimental results is less than two points, suggesting that sampling synthesized demonstrations has a relatively

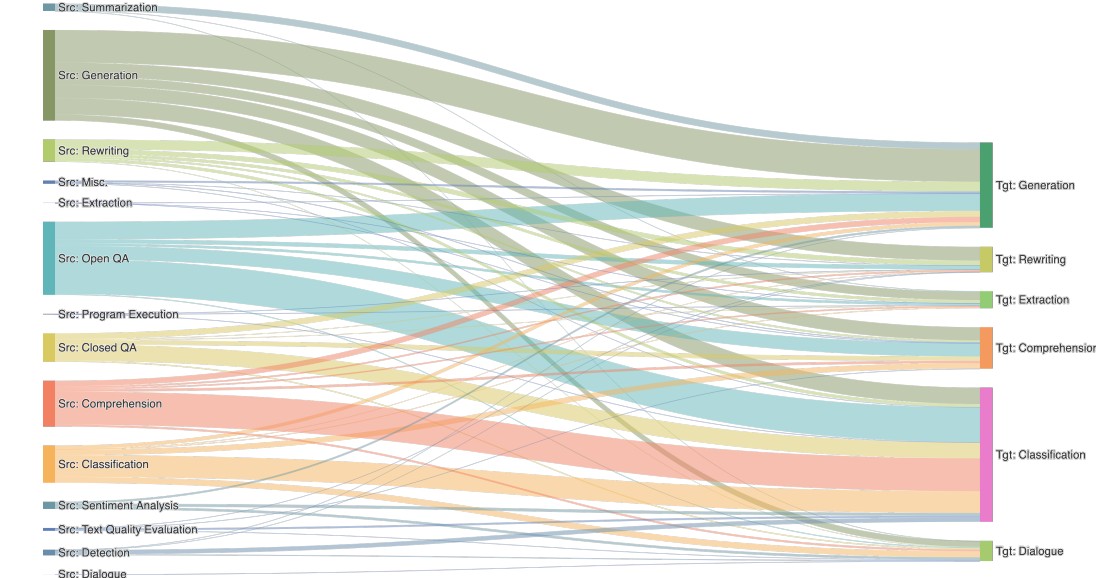

Figure 6: The Sankey figure of the transfer between different source categories and target categories. The category follows Wang et al. (2024b).

small impact on performance, matching the results in Table 2, since we also select the question-related demonstrations during subsequent inference, which overlaps the effectiveness of the target sampling.

### F.3 THE TRANSFER BETWEEN SOURCE DEMONSTRATIONS AND TARGET DEMONSTRATIONS

To investigate the relationships between different source tasks and target tasks, we static the number of target demonstrations corresponding to various source tasks after source sampling. The statistical results are shown in Figure 6, from which we can see that: *(i)* Almost all source tasks contribute to various target tasks, showing the necessity of sampling source demonstrations from different source tasks; *(ii)* Among all source tasks, *Generation* provides significant assistance to all categories of target tasks, indicating that the generalization capability of different source tasks to various target tasks differs; *(iii)* All categories of source tasks contribute demonstrations to target tasks for transfer learning, suggesting that when choosing source tasks to be sampled, it is essential to cover different categories of tasks to help target tasks acquire diverse capabilities and domain knowledge.

### F.4 PASS RATE OF TRANSFER VERIFICATION

Table 9: The pass rate of the transfer verification of ICTL on the Super-NI test set using Llama3.1-8b.

| Category | Pass Rate (%) |
|---|---|
| Classification | 85.6 |
| Comprehension | 68.4 |
| Dialogue | 76.5 |
| Extraction | 80.8 |
| Generation | 61.3 |
| Rewriting | 66.6 |
| Overall | 74.1 |

To verify the quality of the transfer results across different target tasks, we report the pass rates of transfer verification across various task categories, as shown in Table 9. From the table, we can observe that: *(i)* For all task categories, the synthesized demonstrations of ICTL achieve a pass rate

of over 60%, indicating that the synthesized results generally satisfy the requirements of the target tasks; *(ii)* Compared to tasks with more definite answers (e.g., Classification, Extraction), tasks with more open-ended answers (e.g., Generation, Rewriting) exhibit lower pass rates, since during transfer for these tasks, the model struggles to determine the appropriate answer format based on the task definition, leading to poorer transfer results.

### F.5 COMBINE ICTL WITH HUMAN-LABELING DEMONSTRATIONS

Table 10: The performance of ICTL with and without additional human labeling using Llama3.1-8b. **Single** denotes only using the example of each target task. **Multiple** denotes using additional human-labeled demonstrations provided by Super-NI.

| Metric | Single | + ICTL | Multiple | + ICTL |
|---|---|---|---|---|
| EM | 39.7 | 44.0 | 41.5 | 45.6 |
| RougeL | 54.7 | 60.3 | 57.6 | 60.4 |

To verify the performance of our method in the presence of human-labeled demonstrations, we conduct experiments using additional demonstrations labeled by humans. For each test task, we utilize the dataset excluding the 100 test instances as the demonstration pool for the experiments. We perform two sets of experiments: one using only human-labeled demonstrations and the other combined with the demonstrations transferred by ICTL. The experimental results are shown in Table 10. From the table, we can see that compared to the results using only human-labeled demonstrations, our method achieves further performance improvements, demonstrating the effectiveness in augmenting demonstrations labeled by humans.

### F.6 PERFORMANCE OF ICTL CROSS DIFFERENT DOMAIN

Table 11: The cross-domain performance of ICTL on BOSS (Yuan et al., 2023) under different settings present in §4.1.4 using Llama3.1-8b. The performance of each category is evaluated with RougeL. We delete all toxic detection questions because the security restrictions of the model we use lead to refusal to answer questions with sensitive words. The best performance of each category is marked in **bold**.

| Category | Zero | Direct | Single | Synthesis | Ours |
|---|---|---|---|---|---|
| Name Entity Recognition | 28.2 | 84.4 | 85.0 | 84.6 | **85.4** |
| Natural Language Inference | 21.1 | 21.7 | 21.0 | 22.5 | **24.8** |
| Question Answering | 60.6 | 62.5 | 64.2 | 62.3 | **64.8** |
| Sentiment Analysis | 71.5 | 73.8 | 70.0 | 70.8 | **74.0** |
| Overall (EM) | 33.2 | 36.8 | 34.8 | 35.3 | **39.9** |
| Overall (RougeL) | 45.4 | 60.6 | 60.0 | 60.0 | **62.2** |

To evaluate the performance of ICTL across different domains for the same task, we conduct cross-domain experiments. Since all different tasks of Super-NI exhibit some variation, we opt to use BOSS (Yuan et al., 2023) for the experiments, which standardizes the input-output format for data across different domains within the same task, allowing for a more accurate evaluation of cross-domain performance. The experimental results are shown in Table 11, from which we can observe the following: *(i)* Under the setting of the same task across different domains, our method still yields performance improvements, demonstrating its effectiveness in cross-domain scenarios. *(ii)* Apart from our method, *Direct* achieves the best performance, since despite being in different domains, the task and input-output format are identical, allowing the model to learn how to perform accurate reasoning from demonstrations in other domains of the same task.

### F.7 PERFORMANCE OF ICTL WITH SYNTHESIZED DEFINITIONS

Considering that humans could label no task definition in the real application, we discuss the performance of ICTL using the synthesized definitions in this section. We employ Auto-ICL Yang et al.

| Definition | EM | RougeL |
|---|---|---|
| Auto-ICL | 42.3 | 59.1 |
| Human-Labeled | 44.0 | 60.3 |

Table 12: The performance of ICTL using task definitions synthesized by LLMs and labeled by humans on Super-NI.

(2024) to synthesize task definition, where the experiment results are shown in Table 12. From the table, we can find that the performance degradation caused by synthetic definition is not significant. This is because the performance of our method is not particularly sensitive to the similarity between the source task and target task definitions, as shown in Figure 3c.

## G  SYNTHESIS CASE STUDY

Table 13: The case study of the capability transfer for the classification task.

| | | |
|---|---|---|
| **Source Data** | **Definition** | **QA ZRE Question Generation on Subject Relation:** You will be given a context, a subject and a relation. Your task is to generate a question based on the subject and relation. The generated question should include the given subject. Try to use a minimum number of words that are not present in either context, subject or relation while generating question. |
| | **Input** | Context : Blind Company was shot in Bicheno, Tasmania in September 2008. Subject : Blind Company Relation : narrative location |
| | **Output** | Which place is Blind Company in? |
| **Transferred Data** | **Definition** | **Scitail1.1 Classification:** You are given two sentences. You have to find if there is entailment or agreement of the Hypothesis by the Premise. From the given pair of sentences, you should identify if there is enough information in the Premise to support the claim made in the Hypothesis. The Premise may not exactly be the same as Hypothesis. Your task is to return 'entails' if the premise supports hypothesis else return 'neutral'. |
| | **Input** | Premise: Blind Company was shot in Bicheno, Tasmania in September 2008. Hypothesis: Blind Company is in Bicheno. |
| | **Output** | entails |

In this section, we conduct a case study on the data transferred by ICTL to gain a deeper understanding of how task transfer is performed. We investigate from two perspectives: capability transfer (Table 13, Table 15) and domain transfer (Table 14, Table 16). From these cases, we can observe that: *(i)* Capability transfer generally occurs when the source and target tasks are highly similar, where when the definition or format of the source and target tasks are similar, our method can effectively understand the meaning of the source task and apply it to the target task; *(ii)* Domain transfer occurs when there is a significant difference between the source and target tasks, where the model leverages the original input information from the source task, which includes domain knowledge, while the answers or other information for the target task are generated independently by the model.

## H  BAD CASE STUDY

To better understand how ICTL enhances reasoning performance, we analyzed a bad case of Super-NI, as shown in Figure 7. The figure illustrates that without ICTL, the model erroneously focuses

Table 14: The case study of the domain transfer for the classification task.

| | | |
|---|---|---|
| **Source Data** | **Definition** | **XLWIC True or False Answer Generation:**
In this task, you are given a word, followed by two sentences. Your task is to figure out whether both the sentences use the aforementioned word with the same meaning. You should respond with 'True' if the words in both sentences share the same meaning, and 'False' otherwise. |
| | **Input** | spring
Sentence1: I spent my spring holidays in Morocco.
Sentence2: He will hold office until the spring of next year. |
| | **Output** | False |
| **Transferred Data** | **Definition** | **ANLI R2 Entailment:**
In this task, you will be presented with a premise and a hypothesis sentence. Determine whether the hypothesis sentence entails (implies), contradicts (opposes), or is neutral with respect to the given premise. Please answer with "Contradiction", "Neutral", or "Entailment". |
| | **Input** | Premise: The spring season is a time of renewal and growth, often associated with warmer weather and longer days.
Hypothesis: He will hold office until the spring of next year. |
| | **Output** | Neutral |

Table 15: The case study of the capability transfer for the generation task.

| | | |
|---|---|---|
| **Source Data** | **Definition** | **Para-NMT Paraphrasing:**
This is a paraphrasing task. In this task, you're given a sentence and your task is to generate another sentence which express same meaning as the input using different words. |
| | **Input** | someone other than the owner must have known it . |
| | **Output** | someone , outside the owner , must have known about that . |
| **Transferred Data** | **Definition** | **Ollie Sentence Answer Generation:**
Given two noun phrases (arguments) and relationship between them, form a sentence that expresses these arguments with the given relationship. |
| | **Input** | Relationship: 'known'
Argument/Subject 1: 'someone other than the owner'
Argument/Subject 2: 'it' |
| | **Output** | someone other than the owner must have known it. |

on the phrase "`worked fine`", leading to an incorrect answer. However, with ICTL, the model is guided to more comprehensively evaluate the user input, thereby producing the correct result.

Table 16: The case study of the domain transfer for the generation task.

| | | |
|---|---|---|
| **Source Data** | **Definition** | **Peixian Rtgender Sentiment Analysis:** Given a 'poster' sentence and a corresponding 'response' (often, from Facebook or Reddit)classify the sentiment of the given response into four categories: 1) Positive, 2) Negative, 3) Neutral, and 4) Mixed if it contains both positive and negative. |
| | **Input** | Poster: La edad hace de las suyas con mis ojitos. Aging is getting to my eyes. OMG!!!! Responser: sorryy jeje eso dije |
| | **Output** | Neutral |
| **Transferred Data** | **Definition** | **Reddit Tifu Title Summarization:** In this task, you are given a Reddit post as a text. Your task is to generate a title for this text. The title should start with ̈TIFU by ̈, followed by a situation that caused humor. The title should contain 7-12 words, ideally. |
| | **Input** | Text: La edad hace de las suyas con mis ojitos. Aging is getting to my eyes. OMG!!!! |
| | **Output** | TIFU by letting aging ruin my eyes in seconds |

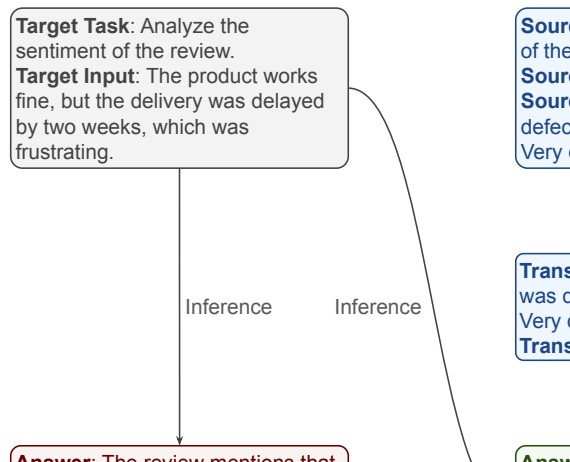

Figure 7: A case of Super-NI without (left) and with (right) ICTL. The correct answer is marked in green and the incorrect answer is marked in red.

