# OpenReview forum: "In-Context Transfer Learning: Demonstration Synthesis by Transferring Similar Tasks"
_ICLR.cc/2025/Conference — Submitted to ICLR 2025_

### Official Review · Reviewer_qFgM · 2024-11-04

**Soundness:** 3
**Presentation:** 3
**Contribution:** 2
**Rating:** 5
**Confidence:** 4

**Summary:**

This paper introduces In-Context Transfer Learning (ICTL), a method to enhance in-context learning by synthesizing examples for new target tasks through the transfer of demonstrations from similar tasks. Unlike conventional synthesis methods constrained by the limitations of LLMs, ICTL leverages labeled demonstrations from related source tasks. The approach involves selecting source tasks closely aligned with the target task and then refining these examples through LLMs to match the target. Experimental results on the Super-NI dataset show that ICTL achieves a 2% average performance improvement over other baselines.

**Strengths:**

- ICTL combines ICL and Transfer Learning, presenting a simple yet effective method for ICL across different tasks, where  target demonstrations are strictly limited.

- ICTL achieves consistent improvement on the Super-NI dataset, with a 2% average performance gain over other baselines.

- The paper is written in a clear manner, making it easy to understand the proposed method and its applications.

**Weaknesses:**

- While the proposed ICTL combines the concepts of Transfer Learning and ICL, its contribution is not clearly specified compared to prior research involving advanced demonstration generation in zero-shot ICL settings and advanced retrieval methodologies for ICL with RAG.

[1] SELF-ICL: Zero-Shot In-Context Learning with Self-Generated Demonstrations (EMNLP 2023)

[2] Demonstration Augmentation for Zero-shot In-context Learning (ACL 2024)

[3] Bridging Distribution Gap via Semantic Rewriting with LLMs to Enhance OOD Robustness (ACL 2024)

- The selected baselines used in the experiments mostly correspond to ablations of the proposed method. For a fairer evaluation, it is necessary to directly compare ICTL with similar approaches discussed in the related work. Specifically, the authors could compare with methodologies that rely on LLMs to generate demonstrations for ICL; e.g., [1, 2] or those that modify source task formats to enhance transfer learning; e.g., [3]. These comparisons could state the contributions and novelty of the proposed ICTL clearly.

- While restructuring the retrieval process based on a Transfer Learning objective is a positive step, the generation of new target demonstrations relies only on LLM prompting and self-verification, which can be insufficient to fundamentally improve task performance. Recent studies such as [4] claim that LLMs are limited in their ability to self-correct reasoning errors. To position target demonstration generation as a significant contribution, the paper needs to further enhance this aspect beyond sampling by proposing novel approaches to optimize the generation process.

[4] LARGE LANGUAGE MODELS CANNOT SELF-CORRECT REASONING YET (ICLR 2024)

- ICTL involves multiple steps, including sampling, transferring, and verifying demonstrations, which may increase computational demands compared to simpler in-context learning approaches. Considering the use of high-cost and high-quality models like GPT-4o and Llama3.1-8b, along with the extra inference required for demonstration generation and self-verification within ICTL, the 2%
average performance improvement seems less significant to justify the increased cost.

- If there are few or no closely matching source tasks, the quality of transferred demonstrations may suffer, impacting the model’s ability to generalize to the target task.

- The Super-NI dataset used to demonstrate performance improvement is somewhat limited. In Section 1, the authors claim that their approach aims to enhance performance on tasks where the LLM lacks prior knowledge. However, it is hard to argue that the tasks or knowledge in the Super-NI dataset are entirely absent from GPT-4o or Llama3.1-8b-Instruct models. To provide more convincing validation, the authors should either use a more up-to-date benchmark or design tasks in environments like Alfworld [5] for additional testing.

[5] ALFWorld: Aligning Text and Embodied Environments for Interactive Learning (ICLR 2021)

**Questions:**

- The experiments in the paper focus solely on final task performance, with no analysis of generated target demonstration quality. It would strengthen the paper’s contributions if more analysis was presented. Could the authors provide example target demonstrations to illustrate the model’s limitations and how they were improved?

- How does ICTL perform when there were no closely related source tasks available? Experimental results on how ICTL performs when there are limited or no closely related source tasks available would be beneficial.

- How does the computational cost of ICTL compare to traditional ICL?

- Could the authors compare ICTL with recent works on learning-based retrievers in LLM RAG approaches instead of Direct?

---

> ### Author Response · Authors · 2024-11-18
> **Responses for Reviewer qFgM**
>
> Thank you for your valuable suggestions and for recognizing our work! Below is our response to your inquiry:
> 1. The contribution is not clearly specified compared to prior research involving advanced demonstration generation in zero-shot ICL settings and advanced retrieval methodologies for ICL with RAG.
>   - Compared with other zero-shot ICL methods:
>     - Our approach is not in competition with other methods for synthetic data generation; instead, it can further improve performance based on these existing synthesis methods.
>     - Experiments in Appendix F.4 demonstrate that, when using existing demonstrations (either human-annotated or LLM-generated), our method can still enhance performance, achieving an average improvement of 4.2%, which confirms its ability to further strengthen performance.
>     - This is because other synthesis methods rely on the capabilities and knowledge inherent to the LLM itself. In contrast, our method leverages knowledge and abilities from other tasks to boost performance, enabling the enhancement of skills or knowledge that the LLM may not inherently possess.
>   - Compared with other retrieval methods:
>     - We compare our method with other retrieval methods during filtering source tasks, where the results are shown in the table
> | Retriever | Direct | ICTL |
> | ---------- | --- | --- |
> | BM25 | 46.2 | 55.8 |
> | Contriever | 46.5 | 56.3 |
> | Dr.ICL | 48.4 | 58.7 |
> | Ours | 48.8 | 60.3 |
>     - From the table, we can see that our method shows higher performance compared with other methods, proving the effectiveness of our method.
>     - We have added the above results to the new version of our paper. (Appendix F.1)
> 2. The self-verification is insufficient to fundamentally improve task performance since the previous works.
>   - The verification step of our method is to verify whether the generated demonstrations match the target task definition, which is different from the Self-Verification methods you mentioned to check the correctness of the generated answer.
>   - We verify whether the generated demonstrations match the target task since unmatched demonstrations could mislead the inference of LLMs.
>   - As shown in Table 2, the verification step of our method can enhance performance indeed.
> 3. The performance improvement seems less significant to justify the increased cost.
>   - The performance improvement of our method is significant:
>     - The performance improvement of our method is related to the task type. As shown in Table 1, on tasks with free-form answers (e.g., Dialogue, Generation), since it is difficult to be completely consistent with the answer, the improvement is not significant; but on tasks with fixed answer forms (e.g., Classification, Comprehension), our method can improve by an average of 4.0%, proving the effectiveness of our method.
>     - Experiments on more specific tasks in Appendix F.6 show that our method can improve EM by 4.6% compared to Synthesis, proving the effectiveness of our method.
>   - The cost increase is not significant:
>     - The demonstration generation is offline, where during the inference, we only need to sample question-related demonstrations from the generation results, having a similar efficiency to the general ICL methods.
> 4. If there are few or no closely matching source tasks, the quality of transferred demonstrations may suffer, impacting the model’s ability to generalize to the target task.
>   - Our source sampling strategy ensures that the source tasks sampled are highly similar to the target tasks.
>   - For tasks that are particularly rare in practical applications, Figure 3(c) shows that our method outperforms the settings without transferring even for the least similar tasks, demonstrating the effectiveness of our method when utilizing dissimilar tasks.
> 5. It is hard to argue that the tasks or knowledge in the Super-NI dataset are absent from GPT-4o or Llama3.1-8b-Instruct models.
>   - Our main experiment demonstrates that our method leads to performance improvements, confirming that the ability required for solving data in Super-NI also extends beyond the capabilities of the original LLM.
>   - Due to limitations in time and computational resources during the rebuttal period, we are unable to conduct experiments on additional datasets. In future work, we plan to evaluate our method on more datasets, including those you suggested (ALFWorld), and incorporate these results into the paper.
> 6. It would strengthen the paper’s contributions if more analysis was presented.
>   - In the new version of our paper, we provide a bad case in Appendix H to better analyze how our method enhances model performance.
>   - In this case, for the sentiment analysis task, we leverage data from the Sentiment-to-Text task to infer the sentiment associated with certain reviews.
>   - Consequently, during sentiment analysis, we can directly determine the sentiment of similar reviews, thereby reducing the inference complexity for the model.

---

> > ### Author Response · Authors · 2024-11-18
> > **Response for Reviewer qFgM**
> >
> > 7. How does the computational cost of ICTL compare to traditional ICL?
> >   - As discussed in the replay of Q3, the demonstration synthesis is offline, so the inference cost is the same as the general ICL methods.
> >   - We also analyze the efficiency of the demonstration generation of ICTL in Appendix E.1.
> >   - We present that we can control and how to control the parameters to ensure the balance between efficiency and effectiveness.

---

> > > ### Comment · Reviewer_qFgM · 2024-11-27
> > >
> > > I appreciate your thoughtful responses and particularly value the additional experiments conducted on the retrieval methods.
> > > I acknowledge that the proposed ICTL introduces some degree of novelty by leveraging LLMs to generate target demonstrations under the assumption that source demonstrations are accessible and the target task is known offline. However, the technical contributions of the two-step process in ICTL, source sampling and target transfer, may not represent a significant advancement over existing approaches. In particular, the target transfer step relies heavily on LLMs and improves transfer performance through a verification process. While this method is effective and widely adopted in LLM reasoning tasks, it is also known to have certain limitations depending on the model and task, as noted in recent studies. Given these limitations and the restricted scope of the experimental evaluations to the specific Super-NI dataset, I maintain my current evaluation score.

---

> > > > ### Author Response · Authors · 2024-11-28
> > > > **Response for the response of Reviewer qFgM**
> > > >
> > > > Sincerely thanks for your response. We understand your concern and here are our responses：
> > > > 1. The verification by LLMs has certain limitations.
> > > > - The verify step of ICTL is **different** from the self-verification methods you mentioned. Our verification step aims to ensure that the generated demonstrations match the target task definition, as mismatched demonstrations could mislead the inference of large language models. Unlike self-verification methods, we focus on the consistency between the demonstrations and the task definition. As shown in Table 2, the verification step in our method does enhance performance.
> > > > 2. The restricted scope of the experimental evaluations to the specific Super-NI dataset
> > > > - Experiments on more datasets (e.g., QA, sentiment analysis) in Appendix F.6 show that our method can improve EM by **4.6%** compared to Synthesis, proving the effectiveness of our method on general tasks.

---

> ### Author Response · Authors · 2024-11-25
> **Kindly Reminder for Reviewer qFgM**
>
> Dear Reviewer qFgM,
>
> As the discussion deadline is approaching (<3 days), we are actively looking forward to your further feedback. Thanks for your effort and understanding!
>
> Kindest regards,
>
> Authors of ICLR Submission 6060

---

### Official Review · Reviewer_Cqhf · 2024-11-04

**Soundness:** 1
**Presentation:** 1
**Contribution:** 2
**Rating:** 3
**Confidence:** 4

**Summary:**

This paper proposes a method to improve ICL (in-context learning) by integrating principles from transfer learning. Specifically, it finds similar demonstrations from another task, using distributional similarity between tasks, to copy into the ICL prompt. It finds very modest increases in ROUGE score on the Super-NI benchmark.

**Strengths:**

1. The proposed idea seems reasonable at a high level.

2. Results show some promise of proposed method.

**Weaknesses:**

I strongly believe this paper, in its current form, does not meet the threshold for ICLR acceptance.

1. It is a bit hard to determine novelty of this work. Authors cite the previous main ICL paradigm of generating synthetic examples from a single human-provided example, but do not compare to prior methods that combine transfer learning, or similarity of examples, to do in-context learning. For instance, Dr.ICL (Demonstration-Retrieved ICL; Luo et al 2023), seems to be retrieving demonstrations that are similar to the task at hand. Thus, Luo et al seems relevant but was not mentioned or compared against.

2. Much of the paper language needs editing. There are issues with writing in most paragraphs.

3. Figure 2 is not particularly informative after Figure 1. Additionally, the examples in Figures 1 and 2 do not make much sense. For instance, in Figure 1, most of the “Definitions” boxes do not make sense (they are not well defined sentences), and the “Question” and “Output” pairs do not make sense either--the output is “Yes” when the question is not a yes/no question. Therefore, it confuses the reader about how the proposed transfer step would help ICL.

4. There are serious coherence issues in Section 3: Method. In Section 3.1: “...We define similarity as, if the sampling scale is N, the N demonstrations most similar to the target task are called similar...” Isn’t this a recursive definition? How are the most similar N demonstrations computed? Variables in 3.1.1 are not well defined, which is a serious issue as this is the only section in the paper with math. What is a hypothesis? What is a task error? What does the “distribution for each task” mean? How are tasks defined statistically/numerically? Where is this task error term used in the method?

5. Evaluations: Table 1 does not seem to show Super-NI to be a challenging task. There is not much of a difference, especially with GPT-4o, on zero-shot and proposed method (68.7 vs 73.1 ROUGE score), as acknowledged by authors. The gap is 52.0 vs 60.3 with Llama3.1-8b. Given the small gap, I would recommend authors to choose a more challenging task where ICL is much more crucial to the model’s success, so that any differences are less likely to be due to noise.

6. Table 2 has a serious typo/weird result. Why is “Source Sample” row, “Rouge” column listed as 43.7 (-3.5), even though 43.7 should be a decrease of -16.6, instead of -3.5? The fact that this number is so low compared to the other entries in the column makes it seem like it is a typo.

7. ROUGE is not necessarily a good metric, depending on the task and dataset answer length. A better one would be expert human evaluations (preferably pairwise evaluations between the methods). If authors do not have the resources for human eval, there are better automated metrics available.

8. All of the results discussion about Figure 3, from Sections 4.4.1 through 4.4.4, lack insight, provide no analysis, and are just restating the graph trends in many words. The redundancy of information throughout this paper is prevalent in other sections as well, not just these sections.

9. Section 4.4.1: “Even transferring using one source demonstration can also effectively improve the performance of the target task. This is because: (i) Even using one single source demonstration, we can also synthesize a large amount demonstrations [sic]  ... (ii) ... even without source demonstrations, LLMs can still synthesize demonstrations...” This is a tangled bunch of sentences that has many issues. First, (ii) seems to contradict the premise of using one source demonstration, as written at the beginning of the section and in (i). Second, (i) and (ii) seem to both talk about the same reason--demonstrations can be synthesized.

10. I encourage the authors to improve their paper by cutting down on a lot of redundancy. A 5-page paper undershooting the page limit is better than a 9-page paper that repeats much of the information multiple times.

**Questions:**

1. What are some examples of source tasks? Where are the 128 or 512 demonstrations (mentioned in section 4.1.5) from? What are source-target task pair examples that were ranked most similar to each other?

2. How many tokens is the average Super-NI answer in your evaluation dataset?

---

> ### Author Response · Authors · 2024-11-18
> **Responses for Reviewer Cqhf**
>
> We sincerely appreciate your insightful feedback and acknowledgment of our efforts. Please find our reply to your question below:
> 1. Authors do not compare to prior methods that combine transfer learning, or similarity of examples, to do in-context learning.
>   - We compare our method with other retrieval methods during filtering source tasks, where the results are shown in the table
> | Retriever | Direct | ICTL |
> | ---------- | --- | --- |
> | BM25 | 46.2 | 55.8 |
> | Contriever | 46.5 | 56.3 |
> | Dr.ICL | 48.4 | 58.7 |
> | Ours | 48.8 | 60.3 |
>   - From the table, we can see that our method shows higher performance compared with other methods, proving the effectiveness of our method.
>   - We have added the above results to the new version of our paper. (Appendix F.1)
> 2. Much of the paper language needs editing.
>   - It would be helpful if you could point out where should be editing.
> 3. Figure 2 is not particularly informative after Figure 1. Additionally, the examples in Figures 1 and 2 do not make much sense.
>   - Figures 1 and 2 are primarily intended to illustrate our motivation and methodology, rather than specific examples, as excessive textual details may distract readers from understanding our method.
>   - For concrete examples, please refer to the case study in Appendix G, where we provide detailed examples of our method.
>   - The "Yes" output in Figure 1 aligns with the task definition, which is to determine whether the question explicitly contains its answer, as described in the blue box and caption of Figure 1.
> 4. There are serious coherence issues in Section 3: Method.
>   - Thank you for your suggestion. This should be expressed as "N source task demonstrations that minimize the target task error after transfer.", where we have updated the corresponding sentences in Lines 158-160.
>   - We have adopted the terminology from [Redko etal](https://arxiv.org/abs/1610.04420), where "Hypothesis" refers to the predictor, "task error" denotes the probability of discrepancy between the predicted and correct results, and "distribution for each task" refers to the data distribution of each task.
>   - In Section 3.2 (Lines 216-219, 227-232), we introduce an Error Term to sample examples in a way that minimizes the overall error, thereby enhancing performance.
> 5. Evaluations: Table 1 does not seem to show Super-NI to be a challenging task.
>   - We adapt our method to more general tasks in Appendix F.6, where our method brings **4.2%** EM improvement, proving the effectiveness of our method.
>   - It would be helpful if you could provide how much improvement is a big gap.
> 6. Table 2 has a serious typo/weird result.
>   - Apologize for our mistake, where the RougeL of -Source Sample should be 56.8.
>   - We have fixed this mistake in the new version of our paper.
> 7. ROUGE is not necessarily a good metric, depending on the task and dataset answer length.
>   - We use RougeL as our evaluation metric because it is employed in the paper of the benchmark of our main experiments (Super-NI).
>   - Additionally, we utilize Exact Match (EM) as a metric, which also demonstrates the effectiveness of our method in enhancing performance.
>   - If you are aware of better evaluation metrics, please let us know, as your suggestions would be highly valuable.
> 8. All of the results discussion about Figure 3, from Sections 4.4.1 through 4.4.4, lack insight
>   - Regarding Figure 3, we not only discuss the trends of the changes but also provide the reasons behind these changes as well as corresponding guidance for subsequent parameter selection.
>   - Any suggestions on how to better discuss the insights would be greatly appreciated.
> 9. Section 4.4.1 is not well present.
>   - This is because we are explaining different phenomena, even if their underlying causes are the same.
> 10. I encourage the authors to improve their paper by cutting down on a lot of redundancy.
>   - It would be very helpful if you could explain which pieces of information are repeated and repeated frequently.
> 11. What are some examples of source tasks? Where are the 128 or 512 demonstrations (mentioned in section 4.1.5) from? What are source-target task pair examples that were ranked most similar to each other?
>   - The source task examples are provided in Appendix G.
>   - The demonstrations for 128 and 512 are synthesized using our method.
>   - We calculated the Wasserstein distance between the source tasks and the target tasks (lines 206-207).
> 12. How many tokens is the average Super-NI answer in your evaluation dataset?
>   - The average number of tokens in the Super-NI evaluation dataset is **5.9**.

---

> ### Author Response · Authors · 2024-11-25
> **Kindly Reminder for Reviewer Cqhf**
>
> Dear Reviewer Cqhf,
>
> As the discussion deadline is approaching (<3 days), we are actively looking forward to your further feedback. Thanks for your effort and understanding!
>
> Kindest regards,
>
> Authors of ICLR Submission 6060

---

> ### Comment · Reviewer_Cqhf · 2024-11-27
> **Response to Author rebuttal**
>
> Thank you for answering my questions and responding to my concerns.
>
> 1) I appreciate the additional baseline you ran (Dr.ICL), and see that there is a small improvement with your method over that baseline.
>
> 2) It will take me many several hours to list out the language changes needed, but as examples:
> - paragraph 2: "For example, a model trained pre-2023 can not use knowledge after 2023, while a model not trained on code tasks can not understand codes well"  should be changed to --> "For example, ... not trained on coding tasks cannot understand code well."
> - contributions section in intro: "We present to synthesize demonstrations by transferring labeled demonstrations of similar tasks, addressing that synthesis from scratch is constrained by the capabilities and knowledge of LLMs;" --> "We argue that answering from scratch is constrained by the capabilities and knowledge of LLMs and thus propose synthesizing demonstrations..."
>
> 3) It is fine to have abbreviated textual examples to not overburden readers, but figures 1 and 2 were very confusing to me as a reader looking at the examples. I understand now why the answer is yes/no when the question is not a yes/no question, after your explanation, but it should be more clear from just reading the figure, or the caption itself.
>
> 4) Thanks for your changes. The similarity definition is an improvement but "In this paper, we define the similarity as: If we want to sample N source demonstrations, the N source task demonstrations can minimize the target task error after transferring." can be more clear. Is similarity a function of two tasks, or of two demonstrations, one from each task?
>
> 5) Presumably you are looking at the zero-to-ours improvement from 60.6 to 64.8%. I would suggest having standard deviations for these results (over multiple LLM calls on the same problem maybe, but this can get expensive), since the absolute percentage is so close to each other between the methods within a task, so we can see how significantly advantageous your results are over baselines.
>
> 6) See below.
>
> 7) I'm not familiar with Super-NI, but if that's the metric that dataset used, then it makes sense why you used it. Especially since the answers are so short (your answer in item 12). I was thinking more of learned metrics between your approach's answers and the ground truth answers, like BERT score or BLEURT, which is in general better than n-gram based approaches like RougeL.
>
> 8) The results section seems more like observations about the graph rather than analysis. For instance in 4.4.1: "(ii) When the sampling scale exceeds 128, there is a slight decrease in performance, indicating that further addition of new source demonstrations does not continue to improve performance, as the number of demonstrations similar to the target task is limited." Why does increasing scale beyond 128 slightly decrease the performance? When using 256 demonstrations, why is additionally passing in 128 less relevant demonstrations, in addition to the more relevant first 128 demonstrations, harmful? Did you try writing in the prompt that the demonstrations are written in order of most to least relevant?
>
> 9) Acknowledged, but should still be written a lot more clearly.
>
> 10) This is mainly a comment about the results section being mostly redundant with the graphs. It is also a blanket statement about being more succinct with each paragraph.
>
> 6, 11-12. Thanks for answering those questions and your fix.
>
> In light of the improvements and explanations, I will slightly improve the score of this paper, but I still do not believe it is close to meeting the threshold for acceptance, mainly because of items 2, 3, 5, 8. Please also include a clearer description of the algorithmic differences between your method and prior retrieval based methods like Dr.ICL for your next paper revision. (ie: don't only show that your method is slightly better, but also provide a hypothesis based on the algorithmic differences for why it does slightly better.)

---

> ### Author Response · Authors · 2024-11-28
> **Response for the Response of Reviewer Cqhf**
>
> Thank you for your helpful suggestions! Considering your main concerns, here are our responses to points 2, 3, 4, and 8:
>
> 2. It will take me several hours to list out the language changes needed.
> - Thank you for your suggestion. We have incorporated your proposed revisions into the new version. However, we remain somewhat puzzled about where improvements in writing are needed, especially since both Reviewer qFgM and Reviewer 6kCu have stated that our paper is well written.
>
> 3. Figures 1 and 2 were very confusing to me as a reader looking at the examples.
> - Thank you for your suggestion. The reason we use this complex example in Figure 1 is that overly simple examples could confuse readers about why LLMs make errors on such straightforward tasks. On the other hand, more complex examples better demonstrate the value of our work.
>
> 5. The absolute percentage is so close to each other between the methods within a task.
>   - The performance improvement of ICTL is related to the task type. As shown in Table 1, on tasks with free-form answers (e.g., Dialogue, Generation), since it is hard to be completely consistent with the answer, the improvement is not significant; but on tasks with fixed answer forms (e.g., Classification, Comprehension), our method can improve by an average of **4.0%**, proving the effectiveness of our method.
>   - Experiments on more specific tasks in Appendix F.6 show that our method can improve EM by **4.6%** compared to Synthesis, proving the effectiveness of our method.
> 8. The results section seems more like observations about the graph rather than analysis.
>
> 8.1. Why does increasing scale beyond 128 slightly decrease the performance?
> - As explained in Section 4.4.1 (ii), the statement "increasing scale beyond 128 slightly decreases performance" is due to "the number of demonstrations similar to the target task is limited."
>
> 8.2. When using 256 demonstrations, why is additionally passing in 128 less relevant demonstrations?
> - Yes, this is because ICTL sample source tasks that are most similar to the target task given a certain scale (as discussed in Section 3.1.2), and then use demonstrations from these source tasks.
>
> 8.3.  Did you try writing in the prompt that the demonstrations are written in order of most to least relevant?
> - As shown in Table 3, our transfer prompt includes only one demonstration, so there is no specific order.

---

### Official Review · Reviewer_6kCu · 2024-11-05

**Soundness:** 3
**Presentation:** 3
**Contribution:** 2
**Rating:** 6
**Confidence:** 3

**Summary:**

This paper proposes In-Context Transfer Learning (ICTL), a framework to enhance demonstration synthesis for in-context learning (ICL) by transferring labeled demonstrations from similar tasks. The motivation is to overcome the limitations of synthesizing demonstrations from scratch using large language models (LLMs). The framework aims to address cost and efficiency issues in generating high-quality task demonstrations by leveraging transfer learning. ICTL has two steps: (1) source sampling: select source tasks similar to the target task by minimizing transfer error. (2) target transfer: transfer these selected demonstrations to the target task’s format using LLMs (transferring, verify, sample). Experiments on the Super-NI dataset show that ICTL achieves 2.0% improvements on average compared to synthesizing demonstrations from scratch.

**Strengths:**

1. The paper is well written and easy to understand. It tackles an important problem of enhancing the generalibility of LLMs.
2. The proposed in-context transfer learning is simple yet produce reasonable improvements over generating demonstrations from scratch. The ablation studies and sensitivity analysis of paramters are beneficial. Evaluations on the Super-NI demonstrate the method’s cross-task performance. The paper also provides a code release for reproducibility, which is beneficial to the community.

**Weaknesses:**

I have two major concerns:

1. The performance improvements seem marginal. The 2% performance improvement over baseline method (generating target demonstartions from scratch) is marginal and may not justify the added complexity of ICTL. The procedure of sampling, transferring, verifying, and re-sampling introduces computational overhead. It would be great to include the computation cost comparison between different approaches in Table 1. Moreover, from the analysis of different parameters in Figure 3, seems the performance can drop signficiantly (more than 2%) if the parameters are not selected properly. This let me suspect the effectiveness of the proposed framework in more general real-world scenarios. It would be interesting to see more results on other tasks.

2. The proposed framework follows the standard transfer learning pipeline. The theoretical analysis is also straight-forward extension. In general, I feel the technical contribution is limited while there are some specific designs for LLM domain.

**Questions:**

While the paper provides an interesting application of transfer learning to in-context learning, I have the concern of the incremental improvement and relatively complex framework. The contribution is not sufficiently novel, and the results are a bit underwhelming. I would be happy if the authors can provide more compelling evidence of ICTL’s practical benefits (e.g., compare the computational cost in Table 1, provide additional evaluation on other tasks).

---

> ### Author Response · Authors · 2024-11-18
> **Responses for Reviewer 6kCu**
>
> Your constructive suggestions and recognition of our work are greatly appreciated. Our response to your query is provided below:
> 1. The performance improvements seem marginal.
>   - The performance improvement of our method is related to the task type. As shown in Table 1, on tasks with free-form answers (e.g., Dialogue, Generation) [[1](https://aclanthology.org/2023.emnlp-main.153/), [2](https://arxiv.org/abs/2406.18365)], since it is difficult to be completely consistent with the answer, the improvement is not significant; but on tasks with fixed answer forms (e.g., Classification, Comprehension), our method can improve by an average of **4.0%**, proving the effectiveness of our method.
>   - Experiments on more specific tasks in Appendix F.6 show that our method can improve EM by **4.6%** compared to Synthesis, proving the effectiveness of our method.
> 2. It would be great to include the computation cost comparison between different approaches in Table 1.
>   - Efficiency during demonstration synthesis
>     - We analyze the efficiency of the demonstration generation of ICTL in Appendix E.1.
>     - We present that we can control and how to control the parameters to ensure the balance between efficiency and effectiveness.
>   - Efficiency during inference:
>     - During inference, the average token number of each question is shown in the table.
> | Method | Zero | Driect | Single | Synthesis | ICTL |
> | --- | --- | --- | --- | --- | --- |
> | Average Tokens | 95.7 | 257.3 | 156.7 | 278.7 | 262.3 |
>     - From the table, we can see that, during inference, the average token number of our method is similar to Direct and Synthesis.
>     - This is because, the demonstration generation is offline, where during the inference, we only need to sample question-related demonstrations from the generation results, having a similar efficiency to the general ICL methods.
>     - We update the above discussion in the new version of our paper. (Appendix E.2)
> 3. The performance can drop significantly (more than 2%) if the parameters are not selected properly.
>   - Our method indeed requires adjusting parameters to achieve optimal performance.
>   - But even if the parameter settings are not optimal, as shown in Figure 3(c), our method still achieves better performance than Synthesis in most parameter settings, proving the effectiveness of our method without specific parameter settings.
> 4. It would be interesting to see more results on other tasks.
>   - We adapt our method to more general tasks (e.g., Question Answering, Sentiment Analysis), where the experiment results are shown in Appendix F.6.
>   - The experimental results show that our method improves EM by **4.6%** compared to Synthesis, proving the effectiveness of our method.
> 5. The technical contribution is limited while there are some specific designs for the LLM domain.
>   - The main topic of our paper is discussing how to synthesize demonstrations for the given task, where the main contributions are as follows:
>     - From the task perspective, past methods synthesize demonstrations with LLMs from scratch, while we propose using demonstrations from other tasks for transferring for the **first** time, which can generate demonstrations beyond the knowledge or capabilities of LLMs themself.
>     - From the method perspective, for the **first** time, we discuss how to sample data similar to the target task (Equation 3) and how to use LLM to migrate examples, and experiments verified the effectiveness of our method.

---

> ### Author Response · Authors · 2024-11-25
> **Kindly Reminder for Reviewer 6kCu**
>
> Dear Reviewer 6kCu,
>
> As the discussion deadline is approaching (<3 days), we are actively looking forward to your further feedback. Thanks for your effort and understanding!
>
> Kindest regards,
>
> Authors of ICLR Submission 6060

---

> > ### Comment · Reviewer_6kCu · 2024-11-27
> > **Response to Authors**
> >
> > Thank the authors for the rebuttal and additional experiments. Most of my concerns are addressed. I would encourage the authors to include the efficiency anlaysis in the main paper. The improvements are somewhat significant (2%-4% on different tasks) although hyperparameter tuning is required. Sometimes the performance drop (>4%) can be larger than the improvements over baselines if the parameters are not optimal. While I am still concerned about the complexity / cost of the proposed pipeline and relative small performance gain, this work presents a valid approach to enhance the generalibility of LLMs. I would like to increase my score to 6.

---

### Official Review · Reviewer_H8mW · 2024-11-09

**Soundness:** 2
**Presentation:** 3
**Contribution:** 3
**Rating:** 6
**Confidence:** 3

**Summary:**

The paper introduces In-Context Transfer Learning (ICTL), a method designed to improve demonstration synthesis for in-context learning (ICL) by leveraging labeled examples from similar tasks.
Unlike conventional approaches that generate task demonstrations from scratch using LLMs, ICTL employs a transfer learning-inspired process consisting of two steps: Source sampling and target transfer.
In the source sampling phase, relevant source tasks are filtered based on task definitions, and demonstrations are subsequently selected using a simulated annealing approach that minimizes transfer error.
During the target transfer phase, LLMs are used to adapt these selected demonstrations to synthesize examples suitable for the target task.
Experimental results on the Super-NI dataset indicate that ICTL outperforms traditional demonstration synthesis methods, effectively addressing the limitations of LLMs in producing high-quality demonstrations from scratch.

**Strengths:**

1. The synthesis of demonstrations from similar tasks is an interesting and novel approach to enhancing the quality of in-context learning.
1. The paper provides an optimization objective for sampling demonstrations, aimed at minimizing the task error bound.
1. The experimental results demonstrate improvements over traditional direct demonstration synthesis methods, showing the effectiveness of the proposed approach.

**Weaknesses:**

1. The proof of Equation 3 (Appendix A) is not clearly explained. At the end of the proof (lines 868-869), the statement "by substituting Theorem 1 into Equation 5, we can derive Equation 3" skips too many intermediate steps, which makes the derivation ambiguous. While Theorem 1 and Equation 5 are related to finding the target task $\hat{\mu}_T$, Equation 3 is focused on determining the source task $\hat{\mu}_S$. This discrepancy is confusing and should be clarified further to convincingly demonstrate that Equation 3 is a reasonable objective.

1. The proposed approach is primarily applicable to benchmarks that provide explicit task definitions, which restricts its applicability to other benchmarks. This limitation confines the evaluation to the Super-NI dataset, and it would be beneficial to explore how this method could generalize to other settings.

1. The sampling process involving simulated annealing lacks sufficient clarity. Given that simulated annealing typically involves multiple iterations, it raises concerns about efficiency. Further details on how the sampling algorithm is implemented, and any measures taken to ensure efficiency, would be helpful.

Other Issues:
1. Rouge -> RougeL

**Questions:**

1. In Equation 2, $\mu$ represents a data distribution of a task, while $x$ is a representation vector of a task definition. Since these two elements are in different spaces, how is the Wasserstein distance between them computed?

1. Lines 206-207 mention that task selection is ranked by the Wasserstein distance between the embedding vectors of task definitions. How is the Wasserstein distance calculated between two vectors? Typically, the Wasserstein distance is used to compare distributions rather than individual points.

1. How does the proposed source task filtering compare to retrieval-based approaches, such as those using Contriever or BERT?

1. What are the "perturbations" used in simulated annealing during the sampling of source demonstrations?

1. What kind of score function is employed in the simulated annealing process?

---

> ### Author Response · Authors · 2024-11-18
> **Responses for Reviewer H8mW**
>
> Sincerely thank you for your recognition of our work and your valuable suggestions! Below is our response to your questions:
> 1. The proof of Equation 3 (Appendix A) is not clearly explained.
>   - We apologize for our mistake, where Equation should be $\hat{\mu}_S = \argmin_{\{\hat{\mu}_{S_i}\}_N} \sum_{i=1}^N \alpha_i W(\hat{\mu}_{S_i}, \hat{\mu}_T)$, where $$\hat{\mu}_T$ is determined by Equation 2.
>     - We apologize that the formula is not well displayed, where you can find the formula in the updated paper.
>   - Then both Equation 3 and Equation 5 are about the source tasks.
>   - We have fixed this mistake in the new version of the paper (Lines 825-828).
> 2. Requiring explicit task definitions could restrict the applicability to other benchmarks.
>   - We can synthesize the task definition based on the demonstrations following [Self-Instruct](https://arxiv.org/pdf/2212.10560) and [Auto-ICL](https://arxiv.org/pdf/2311.09263).
>   - We supply the experiments of our method using the definition synthesized by Auto-ICL, where the EM and RougeL of our method with Llama3.1-8b on Super-NI are shown in the table:
> | Definition | EM | RougeL |
> | --- | --- | --- |
> | Auto-ICL | 42.3 | 59.1 |
> | Human-Labeled | 44.0 | 60.3 |
>   - From the above table, we can find that the performance degradation caused by synthetic definition is not significant. This is because the performance of our method is not particularly sensitive to the similarity between the source task and target task definitions, as Figure (c) shows.
>   - We have added the above experiment to the new version of our paper. (Appendix F.7)
> 3. The sampling process involving simulated annealing lacks sufficient clarity.
>   - The efficiency analysis of simulated annealing is as follows:
>     - Let $t$denote the minimize temperature, $n$ denotes the data scale, and $d$ denotes the dimension of each data.
>     - Then the efficiency of simulated annealing is $\text{O}(-nd^2\log t)$.
>   - Although our method demands the additional cost for computing simulated annealing compared with the general ICL methods, these costs are offline, where our method has the same inference cost as other general ICL methods.
>   - We have updated the above information in the new version of our paper. (Appendix C)
> 4. How is the Wasserstein distance between distribution-point / point-point computed?
>   - We use the same embedding model for the data and task definitions during embedding. Therefore, we can suppose that the task distribution and the task definition vectors are in the same embedding space.
>   - We consider the single point (vector) as the distribution that the variance is 0 to calculate the Wasserstein distance between distribution-point / point-point.
>   - We have added the above information to the new version of our paper. (Lines 184-185)
> 5. How does the proposed source task filtering compare to retrieval-based approaches, such as those using Contriever or BERT?
>   - We compare our method with other retrieval methods during filtering source tasks, where the results are shown in the table
> | Retriever | Direct | ICTL |
> | ---------- | --- | --- |
> | BM25 | 46.2 | 55.8 |
> | Contriever | 46.5 | 56.3 |
> | Dr.ICL | 48.4 | 58.7 |
> | Ours | 48.8 | 60.3 |
>   - From the table, we can see that our method shows higher performance compared with other methods, proving the effectiveness of our method.
>   - We have added the above results to the new version of our paper. (Appendix F.1)
> 6. What are the "perturbations" used in simulated annealing during the sampling of source demonstrations? What kind of score function is employed in the simulated annealing process?
>   - During iteration, perturbation refers to the strategy of randomly replacing one of the currently selected demos with a certain probability, even if the score of the current iteration is not better than the current best score. This approach is applied to avoid getting trapped in a local optimum.
>   - We use Equation 3 as the score function to ensure that the sampled data is relevant to the target task, ensuring performance. We have updated the above information in the new version of the paper. (Appendix C)
> 7. Rouge -> RougeL
>   - Thanks for pointing out this issue. We have changed all of Rouge to RougeL in the new version of our paper.

---

> ### Author Response · Authors · 2024-11-25
> **Kindly Reminder for Reviewer H8mW**
>
> Dear Reviewer H8mW,
>
> As the discussion deadline is approaching (<3 days), we are actively looking forward to your further feedback. Thanks for your effort and understanding!
>
> Kindest regards,
>
> Authors of ICLR Submission 6060

---

> ### Comment · Reviewer_H8mW · 2024-11-25
>
> I raise the score to 6 as most of my concerns are addressed. However, there are some issues regarding the proof, which require further clarification from the authors.
>
> 1. The variable $\hat{\mu}$ in Eq.3 does not appear in the objective. Should it be $\{\hat{\mu}_{S_i}\}$?
>
> 2. What are the difference between
> $\hat{\mu}_{S_i}$ in Eq 3,
> $\hat{\mu}_S$ in Eq.2,
> and $\hat{\mu}_S$
>
> and $\hat{\mu}_{S_i}$ in Eq 5?

---

> > ### Author Response · Authors · 2024-11-26
> > **Responses for Reviewer H8mW**
> >
> > 1. Yes, in Eq.3, $\hat{\mu} =$ { $\hat{\mu}_{S_i}$ }, denoting the distribution of the possible source tasks to be sampled;
> > 2. $ \hat{\mu}_{S_i} $ of Eq.3 and Eq.5 are same, and $ \hat{\mu}_S $ of Eq.2 and Eq.5 are also same:
> > * $ \hat{\mu}_{S_i} $ denotes the distribution of source task $S_i$;
> > * $ \hat{\mu}_S $ denotes the distribution of the sampled source tasks.

---

### Meta-Review · Area_Chair_6rU5 · 2024-12-21

**Metareview:**

The paper presents a transfer learning method which synthesizes target task demonstrations by transferring labeled demonstrations from similar source tasks. The reviewers acknowledged the importance and potential of the problem addressed. However, they identified several limitations that hinder the paper's readiness for acceptance:

- The novelty of the proposed is not sufficiently clear, especially in differentiating it from existing methods, and the contributions do not constitute a substantial advancement.

- The evaluations are limited to the Super-NI dataset, and the performance improvements are modest relative to the computational overhead. Reviewers suggested broader evaluations and deeper insights into the results.

- Some reviewers highlighted issues in writing clarity, methodology explanation, and figure presentation.

- The simulated annealing step and computational costs raised concerns about the method's practicality in real-world scenarios.

Despite the strengths of the paper, such as the reasonably novel combination of ICL and transfer learning, and empirical results demonstrating improvements, the reviewers were not uniformly convinced of its readiness for publication. Therefore, I recommend rejecting this paper in its current form.

**Additional Comments On Reviewer Discussion:**

During the discussion and rebuttal period, reviewers raised concerns about the clarity of the paper, limited novelty of the proposed method, modest performance improvements relative to computational cost, and the restricted evaluation scope. The authors addressed these points by revising the manuscript for clarity, adding comparisons with baselines like Dr.ICL, and providing additional experiments to validate the method’s effectiveness and computational efficiency. While some reviewers acknowledged the authors' efforts and increased their scores, concerns about fundamental novelty and scope persisted.

---

### Decision · Program_Chairs · 2025-01-22

Reject